# Semantically Controllable Generation of Physical Scenes with Explicit Knowledge

## Abstract

Deep Generative Models (DGMs) are known for their superior capability in generating realistic data. Extending purely data-driven approaches, recent specialized DGMs satisfy additional controllable requirements such as embedding a traffic sign in a driving scene by manipulating patterns *implicitly* in the neuron or feature level. In this paper, we introduce a novel method to incorporate domain knowledge *explicitly* in the generation process to achieve the semantically controllable generation of physical scenes. We first categorize our knowledge into two types, the property of objects and the relationship among objects, to be consistent with the composition of natural scenes. We then propose a tree-structured generative model to learn hierarchical scene representation, whose nodes and edges naturally corresponded to the two types of knowledge, respectively. Consequently, explicit knowledge integration enables semantically controllable generation by imposing semantic rules on the properties of nodes and edges in the tree structure. We construct a synthetic example to illustrate the controllability and explainability of our method in a succinct setting. We further extend the synthetic example to realistic environments for autonomous vehicles and conduct extensive experiments: our method efficiently identifies adversarial physical scenes against different state-of-the-art 3D point cloud segmentation models, and satisfies the traffic rules specified as the explicit knowledge.

## 1 Introduction

The recent breakthrough in machine learning enables us to learn complex distributions behind data with sophisticated models. These models help us understand the data generation process so as to realize controllable data generation [1, 65, 16]. Deep Generative Models (DGMs) [23, 34, 17], which approximate the data distribution with neural networks (NN), are representative methods to generate data targeting a specific style, category, or attribute. However, existing controllable generative models focus on manipulating implicit patterns in the neuron or feature level. For instance, [7] dissects DGMs to build the relationship between neurons and generated data, while [55] interpolates in the latent space to obtain vectors that control the poses of objects. One main limitation of these existing models is that they cannot explicitly incorporate unseen semantic rules, which may lead to meaningless generated data that violates common sense. For example, to build diverse physical scenes for evaluating autonomous vehicles, the generated cars should follow semantic traffic rules and physical laws, which cannot be enforced by directly manipulating neurons. In light of the limitations of previous work, we aim to develop a structured generative framework to integrate explicit knowledge [15] during the generation process and thus control the generated scene to be compliant with semantic rules.

Natural scenes can be described with objects and their various relationships [5]. Thus, in this paper, we categorize the semantic knowledge that describes scenes into two types, where the first type denoted as *node-level knowledge* represents the properties of single objects and the second type denoted as *edge-level knowledge* represents the relationship among objects. We also observe that tree structure is highly consistent with this categorization for constructing scenes, where nodes of the tree represent objects and edges the relationship. By automatically controlling the tree structure during the generation, we explicitly integrate the *node-level* and *edge-level* knowledge.

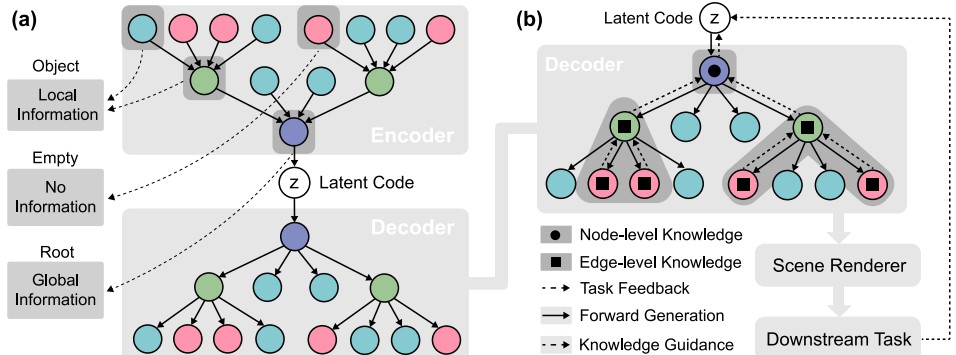

Figure 1: Diagram of proposed SCG. **(a)** *Stage one.* Train T-VAE model to learn the representation of structured data. **(b)** *Stage two.* Integrate node-level and edge-level knowledge during the generation, and generate controllable samples for the downstream task.

In detail, we propose a general framework, Semantically Controllable Generation (SCG), which consists of two stages as shown in Figure 1. In *stage one*, we train a tree-structured generative model that parameterizes nodes and edges of trees with NN to learn the representation of structured data. In *stage two*, explicit knowledge is applied to different levels of the tree to achieve semantically controllable generation for downstream tasks such as satisfying certain conditions or reducing the performance of recognition algorithms.

To verify the proposed SCG, we first construct a synthetic scene reconstruction example to illustrate the advantages of SCG and provide analysis on its controllability and explainability. With SCG, it is possible to generate natural scenes that follow semantic rules, e.g., boxes with the same color should be positioned close to each other. To demonstrate the practicality of SCG, we conduct extensive experiments on adversarial LiDAR scene generation against state-of-the-art 3D segmentation models. We show that our generated safety-critical physical scenes successfully attack victim models and meanwhile follow the specified traffic rules. In addition, compared with traditional attack methods, scenes generated by our method achieve stronger adversarial transferability across different victim models. Our technical contributions are summarized below:

- We propose a semantically controllable generative framework (SCG) via integrating explicit knowledge and categorize the knowledge into two types according to the composition of scenes.
- We propose a tree-structured generative model based on our knowledge categorization and construct a synthetic example to demonstrate the effectiveness of our knowledge integration.
- We propose the *first* semantic adversarial point cloud attack based on SCG, named *Scene Attack*, against state-of-the-art segmentation algorithms, demonstrating several essential properties.

## 2 SEMANTICALLY CONTROLLABLE GENERATION

We define the scene $x \in \mathcal{X}$ in the data space and the latent code $z \in \mathcal{Z}$ in the latent space. This paper aims to generate scene $x$ that satisfies certain semantic rules $\mathbb{K}_t$, which are related to the downstream task $t \in \mathcal{T}$. The scene $x$ will be used to solve the downstream task $t$ by minimizing the corresponding objective function $L_t(x)$. In this section, we first describe the tree-based generative model for learning the hierarchical representations of $x$, which is important and necessary for applying knowledge to achieve semantic controllability. Then we explain the two types of knowledge to be integrated into the generative model together with the generation algorithm that uses explicit knowledge $\mathbb{K}_t$ as guidance for downstream tasks.

### 2.1 TREE-STRUCTURED VARIATIONAL AUTO-ENCODER (T-VAE)

VAE [34] is a powerful DGM that combines auto-encoder and variational inference [9]. It estimates a mapping between data $x$ and latent code $z$ to find the low-dimensional manifold of the data space. The objective function of training VAE is to maximize a lower bound of the log-likelihood of training data, which is so-called Evidence Lower Bound (ELBO)

$$\text{ELBO} = \mathbb{E}_{q(z|x;\phi)}\left[log\, p(x|z;\theta)\right] - \mathbb{KL}(q(z|x;\phi)||p(z)) \tag{1}$$

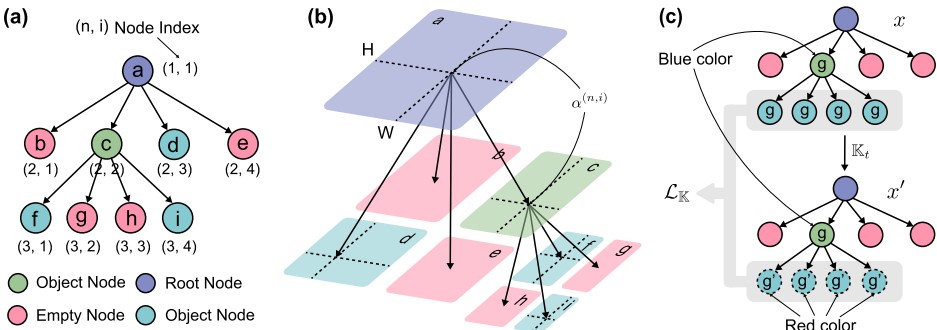

Figure 2: **(a)** Example of tree of a scene with four types of nodes, where the index $(n, i)$ is explained in (2). The purple and green nodes are internal nodes with 4 child nodes. **(b)** The breaking process of a 2D space. The color is corresponding to the node type in **(a)**. **(c)** The knowledge integration example described in Section 2.2: the child nodes of the blue color node should have red color.

where $\mathbb{KL}$ is Kullback–Leibler (KL) divergence. $q(z|x; \phi)$ is the encoder with parameters $\phi$, and $p(x|z; \theta)$ is the decoder with parameters $\theta$. The prior distribution of the latent code $p(z)$ is usually a Gaussian distribution for simplification of KL divergence calculation.

One typical characteristic of natural scenes is the variable data dimension caused by the variable number of objects, which is very challenging to represent with a fixed number of parameters as in traditional models [34]. Besides, many existing structured generative models [64, 40] do not consider the hierarchy of natural scenes nor have the capability to incorporate explicit knowledge.

In general, graph is commonly used to represent structured data [41], but sometimes is too complicated to describe the hierarchy and inefficient to generate. As a special case of graph, tree naturally embed hierarchical information via recursive generation with depth-first-search traversal [29, 48]. This hierarchy is not only highly consistent with natural physical scenes, but also makes easier to apply explicit knowledge, supported by previous works in cognition literature [46].

In this work, we propose a novel tree-structured generative model, which is inspired by the stick-breaking approach [58]: Assume we have a stick with length $W$ and we iteratively break it into segments $w^{(n,i)}$ with:

$$W = w^{(1,1)} = w^{(2,1)} + w^{(2,2)} = \cdots = \sum_{i=1}^{K_n} w^{(n,i)} \tag{2}$$

where $(n, i)$ means the $i$-th segment of the $n$-th level. $K_n$ is the total number of segments in the $n$-th level. The index starts from 1 and entire length has index $(1, 1)$. The recursive function of breaking the stick follows

$$w^{(n+1,j)} = w^{(n,i)}\alpha^{(n,i)}, \quad w^{(n+1,j+1)} = w^{(n,i)}(1 - \alpha^{(n,i)}) \tag{3}$$

where $\alpha^{(n,i)} \in [0, 1]$ is the splitting ratio for $w^{(n,i)}$ and $j$ is the index in the $n + 1$-th level. Intuitively, this breaking process creates a tree structure where the $i$-th level is corresponding to the $i$-th layer of the tree and segments are nodes in the tree. We extent the above division to 2D space with two parameters $\alpha$ and $\beta$. A 2D plane example is illustrated in Figure 2, where each node has 4 child nodes in the next layer.

Assume there are $M$ kinds of nodes in the scene, we define a batch of encoders $E_m$ and decoders $D_m$ for all $m \in M$ and $n \in \{1, \cdots, N-1\}$:

$$f^{(n,j)} = E_m([f^{(n+1,i)}, \cdots, f^{(n+1,i+l_m)}, g^{(n+1,i)}]; \phi_m)$$
$$[f^{(n+1,j)}, \cdots, f^{(n+1,j+l_m)}, \hat{g}^{(n+1,j)}] = D_m(f^{(n,i)}; \theta_m) \tag{4}$$

where $f^{(n,i)}$ is named as the feature vector that passes the messages through the tree structure. $g^{(n,i)}$ is named as the property vector of node $(n, i)$ that stories specific properties such as color of the object generated by node $(n, i)$. $\hat{g}^{(n+1,j)}$ is the predicted property vector and $l_m$ is the number of

children of node $m$. Besides the encoders and decoders, we also define a *Classifier* to determine the child node type and a *Sampler* to infer the posterior distribution of the latent code,

$$\hat{c}^{(n,i)} = \text{Classifier}(f^{(n,i)}; \theta_c), \ \ [z_\mu, z_\sigma] = \text{Sampler}(f^{(1,1)}; \phi_s) \tag{5}$$

where $\hat{c}^{(n,i)}$ is the predicted node type and $[z_\mu, z_\sigma]$ is used to calculate the latent code $z$ with the reparameterization trick [9]. $\theta_c$ and $\theta_s$ are models parameters for the *Classifier* and *Sampler*.

Parameters for the encoders $q(z|x; \phi)$ and decoders $p(x|z; \theta)$ are denoted respectively as $\phi = \{\phi_1, \cdots, \phi_m, \phi_s\}$ and $\theta = \{\theta_1, \cdots, \theta_m, \theta_c\}$. The final structures of the encoder and decoder depend on the tree structure of the data point and vary in the dataset. We follow Recursive Neural Networks (RvNN) [61] to build the tree structure recursively. Finally, the tree $x$ is summarized as

$$x = \{c, g\} = \{c^{(1,1)}, \cdots, c^{(N,K_N)}, \cdots, g^{(1,1)}, \cdots, g^{(N,K_N)}\} \tag{6}$$

where $c$ represents the node type. Following (1), the ELBO of T-VAE to be maximized is

$$\text{ELBO} = \underbrace{\mathbb{E}_q\left[log\, p(c|z; \theta)\right]}_{-\mathcal{L}_C(\hat{c},c)} + \underbrace{\mathbb{E}_q\left[log\, p(g|z; \theta)\right]}_{-\mathcal{L}_R(\hat{g},g)} - \mathbb{KL}\left(\mathcal{N}(z_\mu, z_\sigma) \| \mathcal{N}(0, \mathbf{I})\right) \tag{7}$$

where the equality holds because $c$ and $g$ are conditionally independent given $z$. The $\mathcal{L}_C$ term represents the cross-entropy loss (CE) of all nodes in $p(x|z; \theta)$,

$$\mathcal{L}_C(\hat{c}, c) = \frac{1}{\sum_n^N K_n} \sum_{n=1}^N \sum_{i=1}^{K_n} \text{CE}(\hat{c}^{(n,i)}, c^{(n,i)}; p(c)) \tag{8}$$

where the prior distribution of node type $p(c)$ is calculated from the training dataset and serves as weights. To make the reconstructed tree have same structure as the original one, we use *Teacher Forcing* [67] during the training stage. However, in the generation stage, we take the node with maximum probability as the child to expand the tree. The $\mathcal{L}_R$ term uses mean square error (MSE) to approximate the log-likelihood of node property,

$$\mathcal{L}_R(\hat{g}, g) = \sum_{m=1}^M \frac{1}{N_m} \sum_{n=1}^N \sum_{i=1}^{K_n} \mathbb{1}\left[c^{(n,i)} = m\right] \|\hat{g}^{(n,i)} - g^{(n,i)}\|_2^2 \tag{9}$$

where $N_m$ is the times that node $m$ appears in the tree and $\mathbb{1}[\cdot]$ is the indicator function. In (9), we normalize the MSE with $N_m$ instead of $\sum_n^N K_n$ to avoid the influence caused by imbalanced node type in the tree. Please refer to Appendix B for detailed model definition and generative process.

The advantage of this hierarchical structure is that we only need to store the global information in the root node and use local information in other nodes, making the model easier to capture the feature from different scales in the scene. Moreover, this tree structure makes it possible to explicitly apply semantic knowledge in *Stage 2*, which will be explained in Section 2.2.

## 2.2 KNOWLEDGE-GUIDED GENERATION

Suppose there is a function set $\mathcal{F}$, where the function $f(A) \in \mathcal{F}$ returns true or false for a given input node $A$ of a tree $x$. Then, we define the two types of propositional knowledge $\mathbb{K}_t$ for a particular downstream task $t$ (we omit $t$ below for simplification) using the first-order logic [60] as following.

**Definition 1** *(**Knowledge Set**) The node-level knowledge $k_n$ is denoted as $f(A)$ for a function $f \in \mathcal{F}$, where $A$ is a single node. The edge-level knowledge $k_e$ is denoted as $f_1(A) \rightarrow \forall i\, f_2(B_i)$ for two functions $f_1, f_2 \in \mathcal{F}$, where we apply knowledge $f_2$ to all $A$'s child nodes $B_i$. Then, The knowledge set is constructed as $\mathbb{K} = \{k_n^{(1)}, \cdots, k_e^{(1)}, \cdots\}$.*

In the tree context, $k_n$ describes the properties of a single node, and $k_e$ describes the relationship between the parent node and its children. Thus, the implementation of knowledge is based on the recursive traversal of the tree. We summarized this process in **Algorithm 1**. For the specific knowledge used in our experiment, please refer to Appendix C.

---

**Algorithm 2:** SCG Framework

---

**Input:** Dataset $\mathcal{D}$, Task loss $\mathcal{L}_t(x)$, Searching budget $B$, Knowledge set $\mathbb{K}$
**Output:** Generated scene $x_s$

1  **Stage 1:** Train T-VAE
2    Initialize model parameters $\{\boldsymbol{\theta}, \boldsymbol{\phi}\}$
3    **for** $x$ in $\mathcal{D}$ **do**
4     Encode $z \leftarrow q(z|x; \boldsymbol{\phi})$
5     Decode $\hat{x} \leftarrow p(x|z; \boldsymbol{\theta})$
6     Update parameters $\{\boldsymbol{\theta}, \boldsymbol{\phi}\}$ by maximizing ELBO (7)
7  Store the learned decoder $p(x|z; \boldsymbol{\theta})$

8  **Stage 2:** Generation
9    Initialize latent code $z \sim \mathcal{N}(0, \mathbf{I})$
10   **while** $B$ *is not used up* **do**
11    **if** $L_t(x)$ *is differentiable* **then**
12     $z \leftarrow z - \eta \nabla L_t(p(x|z; \boldsymbol{\theta}))$
13    **else**
14     $z \leftarrow$ Black-box optimization
15    $x' \leftarrow$ ApplyK($\mathbb{K}$, $x$)
16    $z \leftarrow \mathbf{prox}_{\mathcal{L}_{\mathbb{K}}}(z)$ with (10)
17 Decode the scene $x_s = p(x|z; \boldsymbol{\theta})$

---

**Algorithm 1:** Apply Knowledge

---

**Input:** Knowledge $\mathbb{K}$, Tree (root) $x$
**Output:** Modified Tree $x'$

1  **Function** ApplyK($\mathbb{K}$, $x$)
2    **for** *each knowledge* $k^{(n)} \in \mathbb{K}$ **do**
3     $x' \leftarrow$ modify $x$ according to $k^{(n)}$
4    **if** *x has child nodes* **then**
5     **for** *all child nodes* $x_i$ *of* $x$ **do**
6      $x_i' \leftarrow$ ApplyK($\mathbb{K}$, $x_i$)
7      Add node $x_i'$ as a child to $x'$
8  **return** $x'$

---

Then, we apply explicit knowledge to the decoder of T-VAE using the proximal gradient algorithm. The original task objective $\mathcal{L}_t(x)$ is augmented to $\mathcal{L}_a(x) = \mathcal{L}_t(x) + \mathcal{L}_{\mathbb{K}}(x, x')$, where the second term measures the distance between the original tree $x$ and modified tree $x'$. Specifically, $k_n$ changes the property vector $g$ of node $A$ in $x$ to $g'$ to satisfy $f(A)$, and $k_e$ traverses the tree $x$ to find node $A$ that satisfy $f_1$ in $k_e$ then change the type vector $c$ or the property vector $g$ of the children of $A$ to $c'$ and $g'$ respectively. The reference vector $c'$ and $g'$ are related to the downstream task and pre-defined in the function $f$. Then $\mathcal{L}_{\mathbb{K}}$ consists of two parts, MSE is used between the property vector $g$ and $g'$, and CE is used between the type vector $c$ and $c'$.

For instance, the explicit knowledge described as *"if one box is blue, its child nodes should be red"* will be implemented by the following operations. Starting from the root, we find all box nodes whose colors are blue and collect the property vectors $g$ of its child nodes; then we change $g$ to $g'$, which means the color red. Next, $\mathcal{L}_{\mathbb{K}}$ is obtained by calculating MSE between all $g'$ and the original $g$. This process is illustrated in Figure 2(c).

In most applications, $\mathcal{L}_t(x)$ requires many resources to compute, while $\mathcal{L}_{\mathbb{K}}$ is efficient to evaluate since it only involves the inference of $p(x|z; \boldsymbol{\theta})$. Therefore, we resort to Proximal algorithms [52], a type of optimization method, to restrict the searching step to a finite region. In our setting, the explicit knowledge can be treated as the trusted region to limit and guide the optimization of the downstream task. The knowledge loss and task objective are alternatively optimized under the proximal optimization framework. We define the proximal operator as

$$\mathbf{prox}_{\mathcal{L}_{\mathbb{K}}}(z) = \arg\min_{z'} \left( \mathcal{L}_{\mathbb{K}}(p(x|z'; \boldsymbol{\phi}), x') + \frac{1}{2}\|z - z'\|_2^2 \right) \qquad (10)$$

which projects the candidate $z$ to $z'$ to minimizes the knowledge inconsistency caused by $\mathcal{L}_{\mathbb{K}}$. The second term in (10) is a regularize to make the projected point close to the original point.

This problem can be solved by gradient descent since the decoder $p(x|z; \boldsymbol{\theta})$ of T-VAE is differentiable. After the projection, we need to solve the original optimization problem $\mathcal{L}_t(x)$. We use gradient descent for a differentiable $L_t(x)$ or change to black-box optimization methods [6] when $\mathcal{L}_t(x)$ is non-differentiable. The complete algorithm is summarized in **Algorithm 2**.

## 3 EXPERIMENTS

First, we design a synthetic scene to illustrate the controllability and explainability of the proposed framework. The synthetic physical scene provides a simplified setting to unveil the essence of the knowledge-guided generation. After that, we evaluate the performance of SCG on realistic traffic

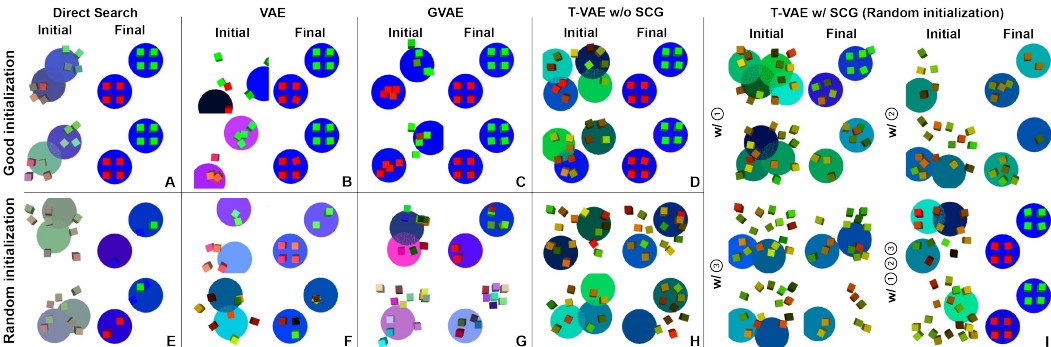

Figure 3: Results of synthetic scene reconstruction experiment from 5 methods with random and good initialization. **I** shows the results of T-VAE using SCG. With the combination of knowledge ①③we can almost reach the optimal solution even from a random initialization, while baseline methods can realize the target *only* when start from the good initialization.

scenes represented by point clouds. Based on SCG, we propose a new adversarial attack method, *Scene Attack*, against multiple state-of-the-art segmentation methods.

## 3.1 SYNTHETIC SCENE RECONSTRUCTION

**Task Description.** We aim to reconstruct a scene to match a given image as shown in Figure 4. The task objective is a reconstruction error $\mathcal{L}_t(x) = \|S - \mathcal{R}(x)\|_2$, where $\mathcal{R}$ is a differentiable image renderer [33] and $S$ is the image of target scene. Under this succinct setting, it is possible to analyze and compare the contribution of explicit knowledge integration, since we can access the optimal solution. According to the understanding of the target scene, we define three knowledge rules: ① *The scene has at most two plates;* ② *The boxes that belong to the same plate should have the same color;* ③ *The boxes belong to the same plate should have distance smaller than a threshold $\gamma$.* We compare our method with three baselines: Direct Search (DS), VAE, and Grammar-VAE (GVAE) [37]. DS directly optimizes the positions and colors of boxes and plates in the data space. In contrast, VAE and GVAE search in the latent space and generate the scene from their decoders.

**Experiment Settings.** We synthesize the dataset by randomly generating 10,000 samples with a varying number of boxes and plates. We also inject the target scene 10 times into the dataset to make sure it accessible to all models. Unlike GVAE and T-VAE, DS and VAE need to access the number of boxes and plates in the target image (e.g., two plates and eight boxes) to fix the dimension of the input feature. To get the good initial points for DS, we add a small perturbation to the positions and colors of all objects in the target scene. Similarly, for other methods, we add the perturbation to the optimal latent code, which is obtained by passing the target scene to the encoder to get good initialization.

| Table 1: Reconstruction Error | | |
|---|---|---|
| | Initialization | |
| Method | Random | Good |
| Direct Search | 86.0±9.4 | **7.9±1.2** |
| VAE [34] | 110.4±10.6 | 13.4±6.1 |
| GVAE [37] | 123.7±9.5 | 19.7±10.2 |
| T-VAE | 135.1±16.9 | 14.1±2.5 |
| T-VAE w/ SCG | **14.5±1.3** | 11.8±2.1 |

**Evaluation Results.** The generated samples from five models are displayed in Figure 3, and the final errors are summarized in Table. 1. With good initialization, all models find a similar scene to the target one, while with the random initialization, all models are trapped in local minimums. However, obtaining good initialization is not practical in most real-world applications, indicating that this task is non-trivial and all models without knowledge cannot solve it. After integrating the knowledge into the T-VAE model, we obtain **I** of Figure 3. We can see that all three knowledge have positive guidance for the optimization, e.g., the boxes concentrate on the centers of plates with knowledge ③ When combining the three knowledge, even from a random initialization, our T-VAE can finally find the target scene, leading to a small error in Table. 1. We also want to mention that it is also possible to apply simple knowledge to GVAE during the generation. However, the advantages of our method are that (1) we can integrate any constraints as long as they can be represented by 1, whereas GVAE can only apply hard constraints to objects with co-occurrences.

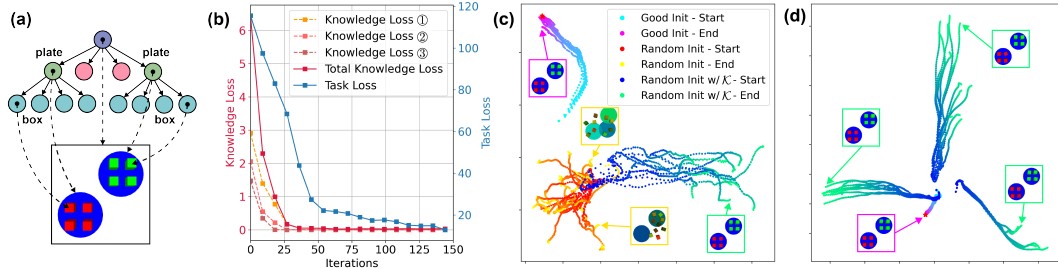

Figure 4: **(a)** Target scene. **(b)** Knowledge losses of integrating semantic rules ①③separately. **(c)** The influence of knowledge to the trajectories with same initialization. **(d)** After applying explicit knowledge, optimization trajectories are diverse when start from different initialization.

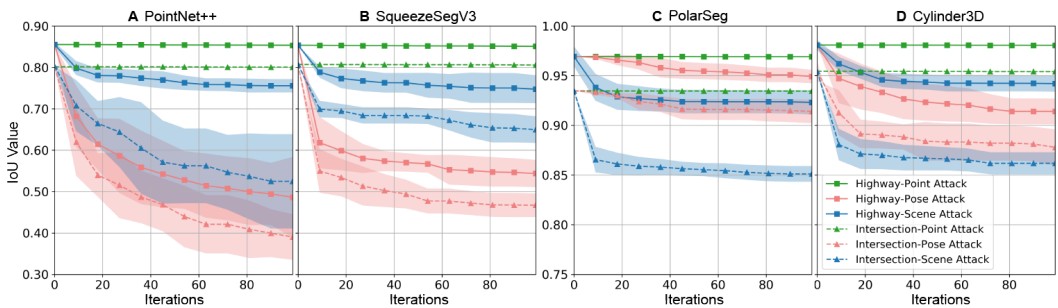

Figure 5: The IoU values during the attack process of four victim models on two backgrounds. Both *Pose Attack* and *Scene Attack* successfully attack all victims, and in some cases, *Pose Attack* outperforms *Scene Attack* in terms of the IoU value. However, scenes generated by *Pose Attack* are not realistic and violate basic physical rules (shown in Figure 6).

**Analysis of Knowledge and Controllability.** To analyze the contribution of each knowledge, we plot the knowledge losses of ①③in Figure 4(b) together with the task loss. All knowledge losses decrease quickly at the beginning and guide the searching in the downstream task. Next, we made ablation studies to explore the reason why knowledge helps the generation. In Figure 4(b), we compare the optimization trajectories of T-VAE (red→yellow) and T-VAE w/ SCG (blue→green) with the same initialization. For T-VAE, the generated samples are diverse but totally different from the target scene, while for T-VAE w/ SCG, the trajectories go to another direction and the generated samples are good. However, the interesting thing is that although the knowledge help us find good scenes, it does not reach the same point in the latent space with the trajectories from good initialization (cyan→purple). This can be explained by the entanglement of the latent space [43], which makes multiple variables control the same property. To further study this point, we plot Figure 4(c), where we use 3 different initialization for T-VAE w/ SCG. The result shows that all three cases find the target scene but with totally different trajectories, which provides evidence to our conjecture. ***In summary, we believe the contribution of knowledge can be attribute to the entanglement of the latent space, which makes the searching easily escape the local minimum and find the nearest optimal points.***

## 3.2 ADVERSARIAL TRAFFIC SCENES GENERATION

**Task Description.** We aim to generate *realistic* traffic scenes against segmentation algorithms as well as satisfying certain semantic knowledge rules. The adversarial scenes are defined as scenes that reduces the performance of downstream victim systems. To generated adversarial LiDAR scenes containing various fore-/background rather than the point cloud of a single 3D object as existing studies [38, 63], a couple of challenges should be considered: First, LiDAR scenes with millions of points are hard to be directly operated; Second, generated scenes need to be realistic and follow traffic rules. Since there are no existing methods to compare with directly, we compare three methods: (1) *Point Attack*: a point-wise attack baseline [69] that adds small disturbance to points; (2) *Pose Attack*: a scene generation method developed by us that searches pose of vehicles in the scene; (3) *Scene Attack*: a semantically controllable generative method based on our T-VAE and SCG. We explore

Table 2: Transferability of Adversarial Scenes (Point Attack IoU / Scene Attack IoU). *Scene Attack* has lower IoU for all evaluation pairs, which demonstrates its better adversarial transferability.

| Source \ Target | PointNet++ | SqueezeSegV3 | PolarSeg | Cylinder3D |
|---|---|---|---|---|
| PointNet++[*] | - / - | 0.916 / **0.768** | 0.936 / **0.854** | 0.955 / **0.918** |
| SqueezeSegV3 | 0.954 / **0.606** | - / - | 0.932 / **0.855** | 0.956 / **0.892** |
| PolarSeg | 0.952 / **0.528** | 0.904 / **0.753** | - / - | 0.953 / **0.908** |
| Cylinder3D | 0.951 / **0.507** | 0.903 / **0.688** | 0.934 / **0.877** | - / - |

[*] For fair comparison, we run 20,000 iterations to get the IoU for *Point Attack.*

the attack effectiveness against different models of these methods, as well as their transferability. For *Pose Attack* and *Scene Attack*, we implement an efficient LiDAR model $\mathcal{R}(x, B)$ [49] (refer to Appendix A for details) to convert the generated scene $x$ to a point cloud scene with an background $B$. The task objective $\min \mathcal{L}_t(x) = \max \mathcal{L}_P(\mathcal{R}(x, B))$ is defined by maximizing the loss function $\mathcal{L}_P$ of segmentation algorithms $P$. We design three explicit knowledge rules: ① *roads follow a given layout (location, width, and length);* ② *vehicles on the lane follow the direction of the lane;* ③ *vehicles should gather together but keep a certain distance.* ① ensure generated vehicles follow the layout of the background $B$ and ③makes the scene contain more vehicles.

**Experiment Settings.** We select 4 segmentation algorithms (PointNet++ [57], PolarSeg [74], SqueezeSegV3 [71], Cylinder3D [75]) as our victim models, all of which are pre-trained on Semantic Kitti dataset [8]. We collect two backgrounds $B$ (Highway and Intersection) in Carla simulator [18]. Since it is usually unable to access the parameters of segmentation algorithms, we focus on the black-box attack in this task. *Point Attack* optimizes $\mathcal{L}_t(x)$ with SimBA [25], while *Pose Attack* and *Scene Attack* optimizes $\mathcal{L}_t(x)$ with Bayesian Optimization (BO) [54]. For the training of T-VAE, we build a dataset by extracting the pose information of vehicles together with road and lane information from Argoverse dataset [11].

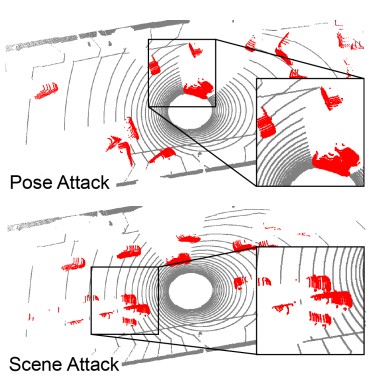

Pose Attack

Scene Attack

Figure 6: Scene generated by *Scene Attack* follows basic traffic rules, while scene generated by *Pose Attack* violates basic physical laws.

**Evaluation Results.** We show the Intersection over Union (IoU) metric for the vehicle during the attack in Figure 5. Generally, it is harder to find adversarial scenes in the highway background than in the intersection background since the latter has much more vehicles. Within 100 iterations, *Point Attack* method nearly has no influence to the performance since it operates in very high dimensions. In contrast, *Pose Attack* and *Scene Attack* efficiently reduce the IoU value. Although *Pose Attack* achieves comparable results to our method, scenes generated by it (shown in Figure 6) are unrealistic due to the overlaps between vehicles. In contrast, scenes generated by our method only modify the vehicles within the traffic constraints. In Table 2, we explore the transferability of *Point Attack* and *Scene Attack*. Transferability, which means using generated samples from the *Source* model to attack other *Target* models, is crucial for evaluating adversarial attack algorithms. Better transferability indicates that the samples carry important patterns that are ignored in most victim systems. Although *Point Attack* method dramatically reduces the performance of all four victims, the generated scenes have weak transferability and cannot decrease the performance of other victim models. However, scenes generated by *Scene Attack* successfully attack all models, even those not targeted during the training, which demonstrates strong adversarial transferability. More generated scenes can be found in Appendix E

## 4 RELATED WORK

**Incorporating Knowledge in Neural Networks.** Integrating knowledge to data-driven models has been explored in various forms from training methods, meta-modeling, embedding to rules used for reasoning. [27] distills logical rules with a teacher-student framework under *Posterior Regularization* [22]. Another way of knowledge distillation is encoding knowledge into vectors then refining the

features from the model that are in line with the encoded knowledge [24]. These methods need to access *Knowledge Graphs* [19] during the training, which heavily depends on human experts. Meta-modeling of complex fluid is integrated into the NN to improve the performance of purely data-driven networks in [45]. In addition, [72] restricted generative models' output to satisfy physical laws expressed by partial differential equations. In the reinforcement learning area, reward shaping [50] is recognized as one technique to incorporate heuristic knowledge to guide the training.

**Structured Deep Generative Models.** The original DGMs, such as GAN [23] and VAE [34], are mostly used for unstructured data. They leverage the powerful feature extraction of neural networks to achieve impressive results [32, 10]. However, the physical world is complex since objects have diverse and structured relationships. Domain specific structured generative models are developed via tree structure (RvNN-VAE [40]) or graph structure (Graph-VAE [59]). Rule-based generative models are also explored by sampling from pre-defined rules [37, 31, 14]. Typical applications of this kind of model are molecular structure generation [26, 29, 30], natural scene generation [51, 13], and automatic program generation [12]. Unlike these existing methods, our approach explicitly integrates knowledge during the generation process. One valuable application of DGMs is generating samples that meet the requirements of downstream tasks [20, 65]. [1, 2] searches in the latent space of StyleGAN [32] to obtain images that are similar to a given image. For structured data, such a searching framework transforms discrete space optimization to continuous space optimization, which was shown to be more efficient [44]. However, it may not guarantee the rationality of generated structured data due to the loss of interpretability and constraints in the latent space [12].

**Physical Scene Generation.** Traditional ways of scene generation focus on sampling from pre-defined rules and grammars, such as probabilistic scene graphs used in [56] and heuristic rules applied in [18]. These methods rely on domain expertise and cannot be easily extensible to large-scale scenes. Recently, data-driven generative models [14, 64, 51, 40, 36] are proposed to learn the distribution of objects and decouple the generation of scene into sequence [64] and graph [51, 40] of objects. Although they reduce the gap between simulation and reality, generated scenes cannot satisfy specific constraints. Another substantial body of literature [21, 35, 24] explored learning scene graphs from images or generating scene images directly via an end-to-end framework. Their generalization to high-dimensional data is very challenging, making them less effective than modularized methods proposed by [36, 68, 14].

**Semantically Adversarial Attacks.** Early adversarial attack methods focused on the pixel-wise attack in the image field, where $L_p$-norm is used to constrain the adversarial perturbation. For the sake of the interpretability of the adversarial samples, recent studies begin to consider *semantic attacks*. They attack the rendering process of images by modifying the light condition [42, 73] or manipulating the position and shape of objects [4, 70, 28]. This paper explores the generation of adversarial point cloud scenes, which already has similar prior works [66, 3, 62]. [66, 3] modify the environment by adding objects on the top of existing vehicles to make them disappear. [62] create a ghost vehicle by adding an ignorable number of points. However, they modify a single object without considering the structural relationship of the whole scene.

## 5 CONCLUSION

In this paper, we explore semantically controllable generation tasks with explicit knowledge integration. Inspired by the categorization of knowledge for the scene description, we design a tree-structured generative model to represent structured data. We show that the two types of knowledge can be explicitly injected into the tree structure to guide and restrict the generation process efficiently and effectively. After considering explicit semantic knowledge, we verify that the generated data contain dramatically fewer semantic constraint violations. Meanwhile, the generated data still maintain the diversity property and follow its original underlying distribution. Although we focus on the scene generation application, the SCG framework can be extended to other structured data generation tasks, such as chemical molecules and programming languages, showing the hierarchical properties. One assumption of this work is that the knowledge is helpful or at least harmless as they are summarized and provided by domain experts, which needs examination in the future.

**Ethics Statement**

Our proposed method aims to enhance existing data-driven generative models with explicit knowledge summarized by human experts. In this paper, we focus on the application of physical scene generation. Considering the broad usage of this method in human society, we identify three potentially harmful insights:

1) The explicit knowledge we integrate is summarized by human experts, therefore, could be biased. With such kind of knowledge, the generated data could also be biased and even results in discrimination.

2) The knowledge we use might potentially contain private information. After integrating it into the generative models, there is a risk of user information disclosure.

3) The knowledge-enhanced generation model is powerful to generate realistic adversarial examples. These examples might be used to attack autonomous systems in the real world, e.g. self-driving algorithms, and cause damages.

To mitigate the potential negative ethics impacts mentioned above, we encourage research to follow these instructions:

1) Construct a strict mechanism of knowledge screening and monitor the generated samples against bias and discrimination.

2) We can also propose a knowledge screening mechanism or an intelligent discriminator to reduce the private information content.

3) Since our generated scenes are realistic and follow traffic rules, it is reasonable to make the defense models drive safely in our generated scenes. However, we should restrict the usage of our generated scenes only in evaluation and testing stages before deployment to help self-driving algorithms overcome their shortcomings.

**Reproducibility Statement**

All of our experiments are conducted on a desktop with Intel i9-9900K CPU @ 3.60GHz, NVIDIA GeForce GTX 1080Ti, and 64GB memory. The settings of the experiment environment are described in the code folder in the supplementary. To make the experiment results reproductable, we provide all pre-trained models (please follow the instruction in *code/README.md* to download). The hyper-parameters of the two experiments are shown in Table 3 and Table 4 in the Appendix.

The dataset we used in *Synthetic Scene Reconstruction Experiment* is synthesized by ourselves, which can be downloaded from the link provided in the *README.md* file. This dataset is published under MIT License. The dataset we used in *LiDAR Scene Generation Experiment* is based on the Argoverse dataset, which is published by Argo AI with MIT License. We use the entire tracking part of the dataset for training and the processed dataset can be downloaded from the link provided in the *README.md* file.

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

## A    LiDAR Model Implementation

The LiDAR model is implemented by Moller-Trumbore algorithm [49] with the PyTorch [53] package for high-efficiency computation. We assume there is a plane $\triangle V_0V_1V_2$ in the 3D space constructed by points $V_0$, $V_1$, $V_2$. A ray $R(t)$ with origin $O$ and normalized direction $D$ (we make $\|D\|_2 = 1$ for simplification) is represented as $R(t) = O + tD$, where $t$ is the distance between $O$ and the end point $D$ of the ray. If $D$ is the intersection between the ray and the plane $\triangle V_0V_1V_2$, we can represent the ray with barycenteric coordinate:

$$T(u, v) = (1 - u - v)V_0 + uV_1 + vV_2 = O + tD = R(t) \tag{11}$$

where $u$ and $v$ are weights. To simplify the equations, we define three new notations:

$$\begin{cases} E_1 = V_1 - V_0 \\ E_2 = V_2 - V_0 \\ T = O - V_0 \end{cases} \tag{12}$$

Then, we can solve the distance $t$ in (11) by:

$$\begin{bmatrix} t \\ u \\ v \end{bmatrix} = \frac{1}{|-D, E_1, E_2|} \begin{bmatrix} |-T, E_1, E_2| \\ |-D, T, E_2| \\ |-D, E_1, T| \end{bmatrix} = \frac{1}{(D \times E_2) \cdot E_1} \begin{bmatrix} (T \times E_1) \cdot E_2 \\ (D \times E_2) \cdot T \\ (T \times E_1) \cdot D \end{bmatrix} \tag{13}$$

Since we know the ray direction $D$, we can get the 3D coordinate of the intersection with this distance $t$. During the implementation, we reuse $D \times E_2$ and $T \times E_1$ to speed up the computation. To make sure the intersecting point $T(u, v)$ is inside the triangle $\triangle V_0V_1V_2$, we need to have:

$$u, v, (1 - u - v) \in [0, 1] \tag{14}$$

If these three conditions are not fulfilled, the intersection point will be removed.

To calculate the point cloud generated by a LiDAR, we first convert vehicle mesh models to triangles $\mathcal{F}$ with Delaunay triangulation [39]. Then, we create the array of LiDAR rays with width $W$ and height $H$ for 360° view. Finally, we use (13) to calculate the intersection point between all triangles $\mathcal{F}$ and LiDAR rays $R_{i,j}(t)$ in parallel to get the final range map $F^{H \times W}$. The background pointcloud will also be converted to range map $B^{H \times W}$ by:

$$\begin{cases} \theta = \arctan \dfrac{z}{\sqrt{x^2 + y^2}} \\ \phi = \arctan \dfrac{x}{y} \\ t = \sqrt{x^2 + y^2 + z^2} \end{cases} \tag{15}$$

where $(x, y, z)$ is the coordinate of one point in pointcloud, $\theta$ is used to calculate the index of row, and $\phi$ is used to calculate the index of column. Then we mix $F^{H \times W}$ and $B^{H \times W}$ by taking the minimal value for each element:

$$M_{i,j} = \min\{F_{i,j}, B_{i,j}\}, \quad \forall i \in W, \ j \in H \tag{16}$$

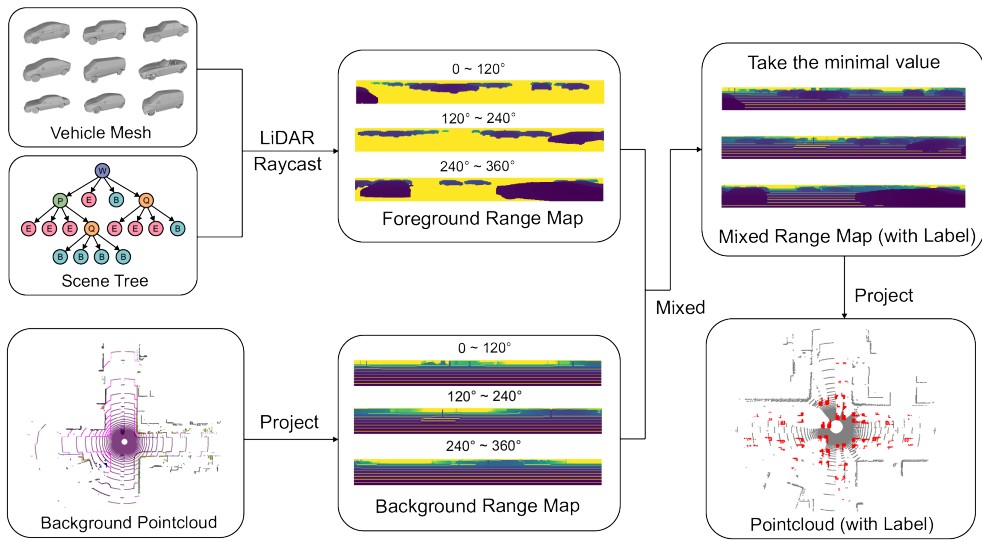

Figure 7: The pipeline of LiDAR scene generation with our developed model.

where $M_{i,j}$ represent the $(i, j)$-entry of the mixed range map scene $M$. Then, we convert the range map to the final output pointcloud scene $S$ with:

$$\begin{cases} z = t \times \sin\theta \\ x = t \times \cos\theta\cos\phi \\ y = t \times \cos\theta\sin\phi \end{cases} \tag{17}$$

The entire pipeline of the above process is summarized in Figure 7. The parameters we used for LiDAR follow the configuration of the Semantic Kitti dataset, where the channel $H = 64$, the horizontal resolution $W = 2048$, the upper angle is $2°$, and the lower angle is $-25°$. The gap between reality and simulation can be reduced by realistic simulation and sensor models [47], but this will not be explored in this paper.

## B  DETAILED T-VAE MODEL STRUCTURE

Our T-VAE model consists of several encoders and decoders that are related to the definition of the scene. In this paper, we explored two experiments with two scenes: a synthetic box placement image scene and a traffic point cloud scene. In Figure 8, we show the details of the modules for two scenes. We also use an synthetic scene as an example to show the encoding and decoding processes in Figure 9, where original data point is stored in tree structure and the generated data is also a tree structure. The encoding process converts the tree into a stack and then encodes the information using the node type defined in the tree. The decoding process expands the tree with the predict node type from the *Classifier*.

For the synthetic box placement image scene, there are 5 *Encoder-Decoder* pairs, a *Sampler*, and a *Classifier*. W node determines the global information such as location, and orientation of the entire scene. P node spawns a plate object in the scene with positions and colors determined by the property vector. Both Q and B nodes spawn a box object in the scene with positions and colors determined by the property vector. E node serves as a stop signal to end the expansion of a branch, therefore, will not spawn anything in the scene and does not have model parameters.

The traffic point cloud scene has similar definitions. R node contains the information of the road and only has two children. L node determines the lane information such as the width and direction. Both Q and V nodes spawn a vehicle in the scene with positions and orientations determined by the property vector.

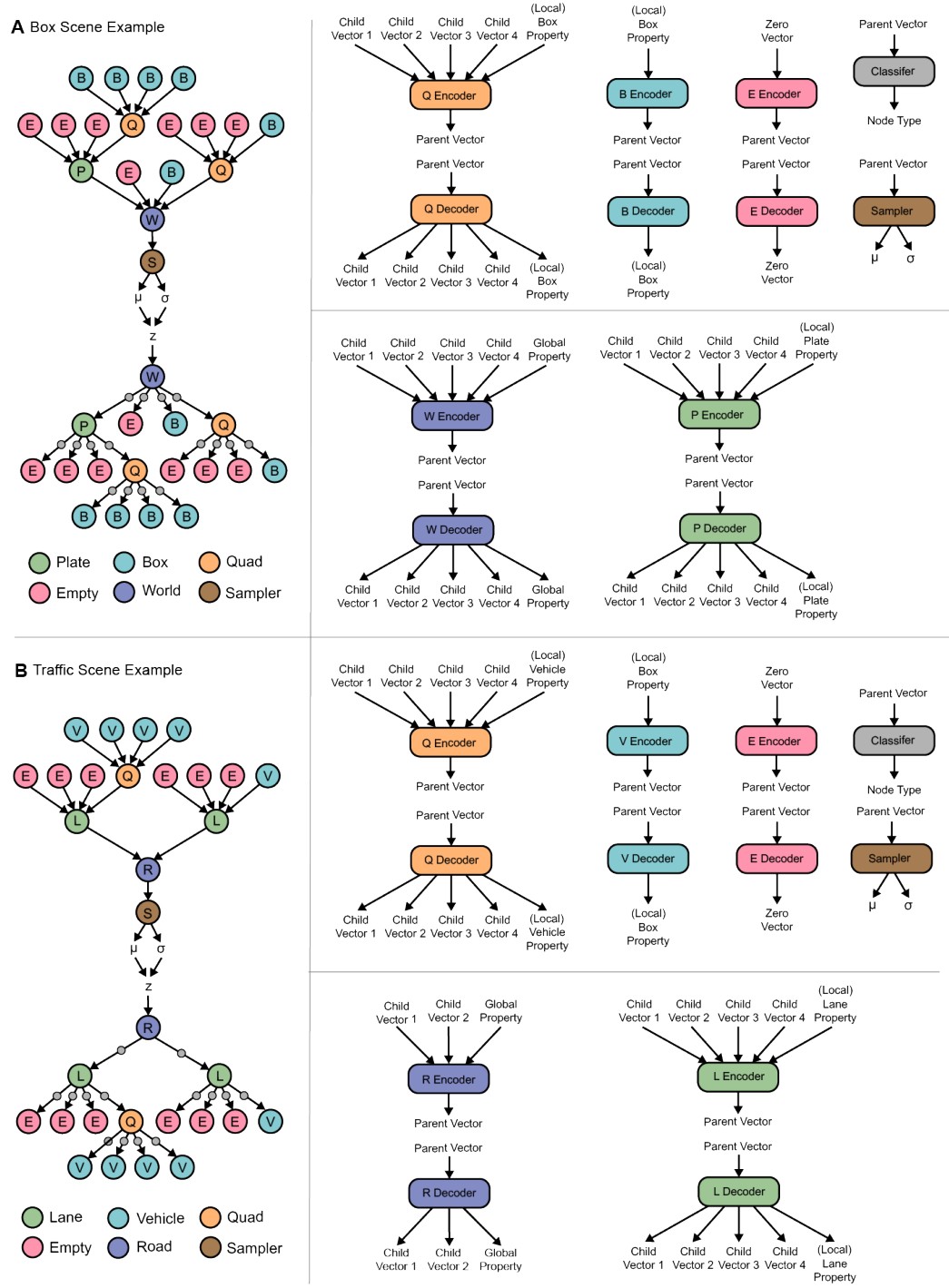

Figure 8: The definition of each module in our proposed T-VAE. **A**: There are 5 kinds of nodes in Synthetic Scene Reconstruction experiment. Therefore, we have 5 encoders and 5 decoders in total, plus a *Classifier* and a *Sampler*. **B**: There are 5 kinds of nodes in the LiDAR scene experiment. Therefore, we have 5 encoders and 5 decoders in total, plus a *Classifier* and a *Sampler*.

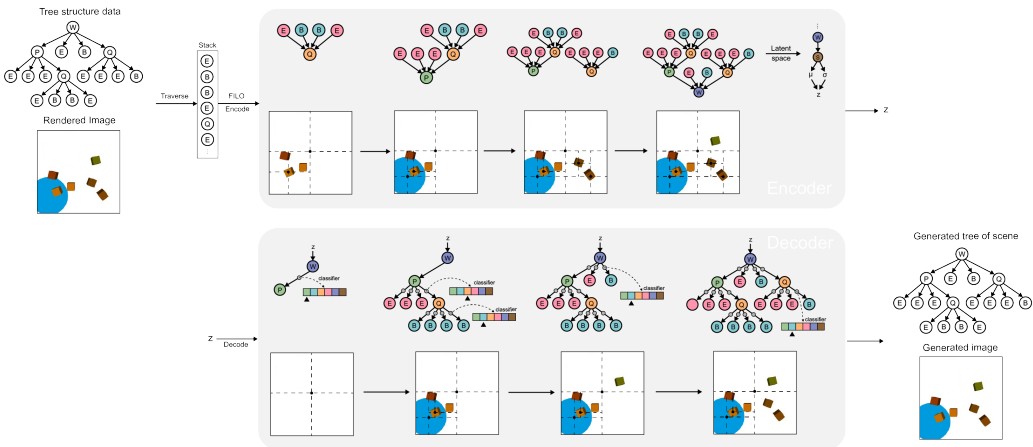

Figure 9: An example of the encoding and decoding processes of our T-VAE model. The circles have the same meaning as in Figure 8.

Table 3: Hyper-parameters of the Synthetic Scene Reconstruction Experiment

| Parameters | Value | Description |
|---|---|---|
| $lr$ | 0.001 | Learning rate of T-VAE training |
| $E$ | 1000 | Maximum training epoch |
| $B$ | 128 | Batch size during training |
| $\eta$ | 0.1 | Learning rate in stage 2. |
| $T$ | 100 | Maximum searching iteration |
| $d_z$ | 64 | Dimension of latent code $z$ |
| $d_f$ | 128 | Dimension of feature vector $f$ |
| $d_g$ | 6 | Dimension of property vector $g$ |
| $\gamma$ | 2 | The threshold used in knowledge ③ |
| $N_l$ | 10 | Normalization factor of location |

Table 4: Hyper-parameters of the LiDAR Scene Generation Experiment

| Parameters | Value | Description |
|---|---|---|
| $lr$ | 0.001 | Learning rate of T-VAE training |
| $\epsilon$ | 0.01 | Max value for point-wise disturbance |
| $E$ | 1000 | Maximum training epoch |
| $B$ | 128 | Batch size during training |
| $T$ | 100 | Maximum searching iteration |
| $d_z$ | 32 | dimension of latent code $z$ |
| $d_f$ | 64 | dimension of feature vector $f$ |
| $d_g$ | 3 | dimension of property vector $g$ |
| $N_l$ | 40 | Normalization factor of location |
| $(w, h)$ | (1.5, 3) | The thresholds used in knowledge ③ |

## C   KNOWLEDGE DEFINITION

For each experiment in this paper, we design three knowledge rules. We explain the details of the implementation of these rules.

In the Synthetic Scene Reconstruction Experiment, we calculate $\mathcal{L}_Y(x, Y_t(x))$ with the following implementations:

① *The scene has at most two plates*, which can be implemented by $W$ *node has at most two* $P$ *children nodes*. We traverse the entire generated tree $x$ to find $W$ node, then we collect the children nodes of $W$ and count the number of $P$ node. If the number is larger than 2, we calculate the cross-entropy loss between the node type and $E$ node label.

② *The colors of the boxes that belong to the same plate should be the same*, which can be implemented by *The color of* $P$ *node's children nodes should be the same*. We traverse the entire generated tree $x$ to find all $P$ nodes, then we collect the colors of the children of $P$. The average color $\bar{c}$ is calculated for each $P$ node and $\bar{c}$ is used as label to calculate the MSE for all children nodes of the corresponding $P$ node.

③ *The distance between the boxes that belong to the same plate should be smaller than a threshold* $\gamma$ which can be implemented by *The distance between* $P$ *node's children nodes should be smaller than a threshold* $\gamma$. We traverse the entire generated tree $x$ to find $P$ node, then we collect the absolute position of all its children nodes and calculate the MSE between this position and the position of $P$.

In the LiDAR Scene Generation Experiment, we calculate $\mathcal{L}_Y(x, Y_t(x))$ with the following implementations:

① *Roads follow a given layout (location, width, and length)*, which can implemented by $R$ *node follows a given layout (location, width, and length)*. We traverse the entire generated tree $x$ to find $R$ node, then calculate the MSE between the property vector of $R$ and the given layout.

② *Vehicles on the lane follow the direction of the lane*, which can implemented by $L$ *node follows pre-defined directions*. We traverse the entire generated tree $x$ to find $L$ node, then calculate the MSE between the property vector of $R$ and the given layout.

③ *Vehicles should gather together but keep a certain distance*, which can implemented by $Q$ *node has at least two* $Q$ *nodes as its children until the absolute width and height of current block is smaller than thresholds* $w$ *and* $h$. We traverse the entire generated tree $x$ to find $Q$ node, then collect the type of its children nodes. When the collected $Q$ node type is less than 2, we calculate the cross-entropy loss between two collected node type and $Q$ node label. When the width and height of current block is smaller than $w$ and $h$, we stop applying this rule.

After calculating all errors in $\mathcal{L}_Y(x, Y_t(x))$, we can directly use back-prorogation to calculate the gradient of latent code $z$ and update it with gradient descent method.

## D   BASELINES IN SYNTHETIC SCENE RECONSTRUCTION

### D.1   DIRECT SEARCH

The dimension of the physical space is $6 \times (2 + 8) = 60$, where $6$ is the property dimension including position, orientation, and colors, $2$ is the number of plates, and $8$ is the number of boxes. We use gradient descent with the learning rate $\eta$ to directly search in the physical space. Since there is no constraint to avoid overlaps between boxes, the generated scenes could be unrealistic.

### D.2   VARIATIONAL AUTO-ENCODER (VAE)

The input dimension of the encoder is the same as the searching space of *Direct Search*. The VAE model has a encoder and a decoder, both of which has a fixed number of model parameters. There are 4 hidden layers in the encoder and each layer has $128$ neurons. The decoder also has 4 hidden layers with $128$ neurons. For the output of color, we add a $Sigmoid(\cdot)$ function to normalize it to the range $[0, 1]$. The location is normalized by $N_l$ before it is taken into the encoder.

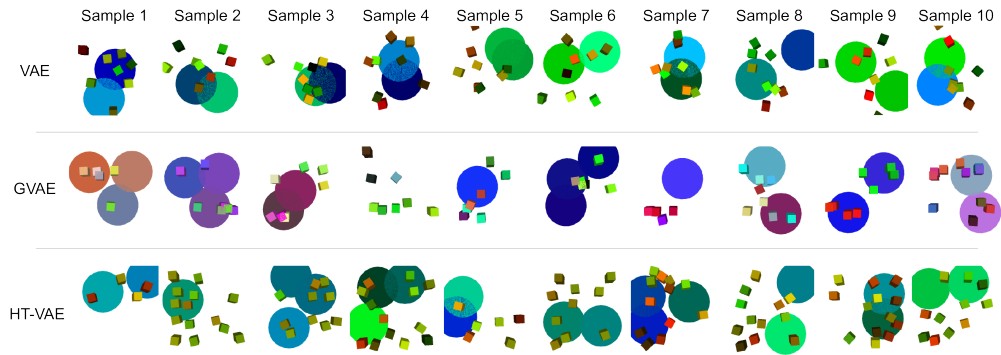

Figure 10: We randomly generate 10 samples by sampling in the latent space of VAE, GVAE, and T-VAE.

### D.3 GRAMMAR VAE (GVAE)

GVAE [37] requires the input data to be described with a set of pre-defined grammars. According to the task of synthetic scene reconstruction experiment, we design 9 rules,

$$W \rightarrow P, W \rightarrow B, P \rightarrow P|E, P \rightarrow P|B, P \rightarrow B, P \rightarrow E, B \rightarrow B, B \rightarrow E, E \rightarrow E \quad (18)$$

and they are represented in one-hot vector in the dataset. The original GVAE is designed only for rule-based discrete data generation (e.g. molecules), thus we modify the structure to add continuous attribute representation. The encoder consists of 3 1-dimensional Convolution layers with kernel size $3 \times 3$. The numbers of channels for the Convolution layers are $[32, 64, 128]$. The decoder is an LSTM model with 128 neurons, therefore, the decoding process is sequential. During the training stage, the maximal length of rules is fixed to 20 with $E \rightarrow E$ as padding, and the decoder will output 20 rules. The cross-entropy loss is used between the input rules and decoded rules. During the generation stage, the rules generated from the decoder will be firstly stored in a stack and convert to the tree with the first-in-last-out (FILO) principle.

## E  ADDITIONAL QUALITATIVE RESULTS

In Figure 10, we show samples randomly generated from VAE, GVAE, and T-VAE. The results show that all three models are able to generate diverse samples. Specifically, samples generated by VAE always have 2 plates and 8 boxes due to the fixed input data dimension. In contrast, samples from GVAE and T-VAE have variable numbers of plates and boxes.

In Figure 11, we show 4 more generated scenarios from the *Point Attack* methods with prediction results from 4 segmentation models. For better visualization, we also show three detailed figures. In Figure 12 and Figure 13, we show more results from *Pose Attack* and *Scene Attack* methods. Scenes generated by *Scene Attack* follow basic traffic rules.

## F  SEGMENTATION MODELS IN LiDAR SCENE GENERATION

### F.1  POINTNET++

This model [57] directly uses point-wise features in the 3D space as the backbone to deal with the segmentation problem. Although this model does not have impressive results on the Semantic KITTI dataset, we select it because it influences a lot of existing point cloud processing models. We use the code from this repository and train the model on Semantic KITTI dataset by ourselves following the original training and testing split setting.

### F.2  POLARSEG

This model [74] converts the data representation from 3D Cartesian coordinate to Polar coordinate and extracts features with 2D convolution layers. We use the code from this repository and use the

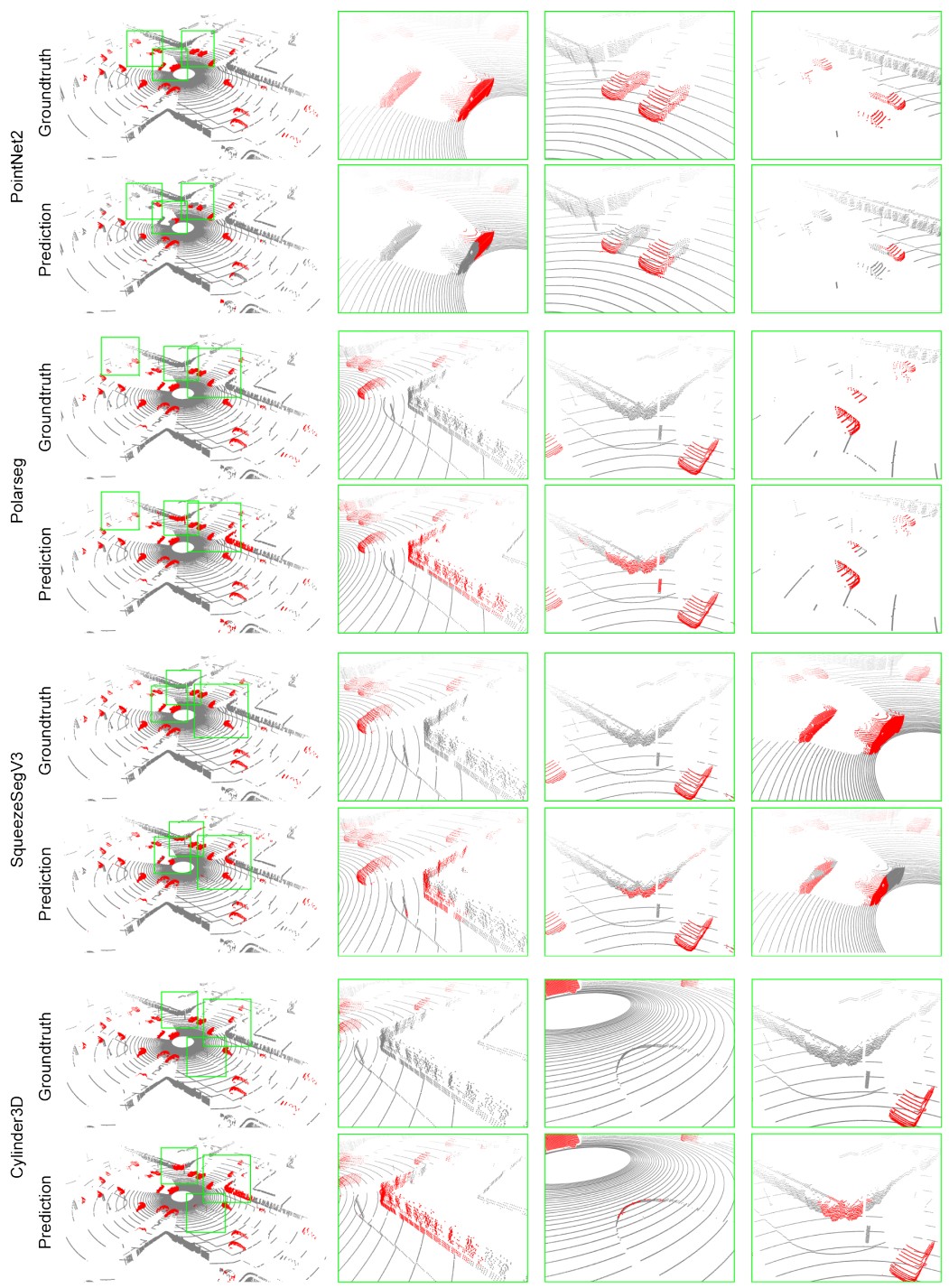

Figure 11: More results for *Point Attack* method with the *Intersection* background.

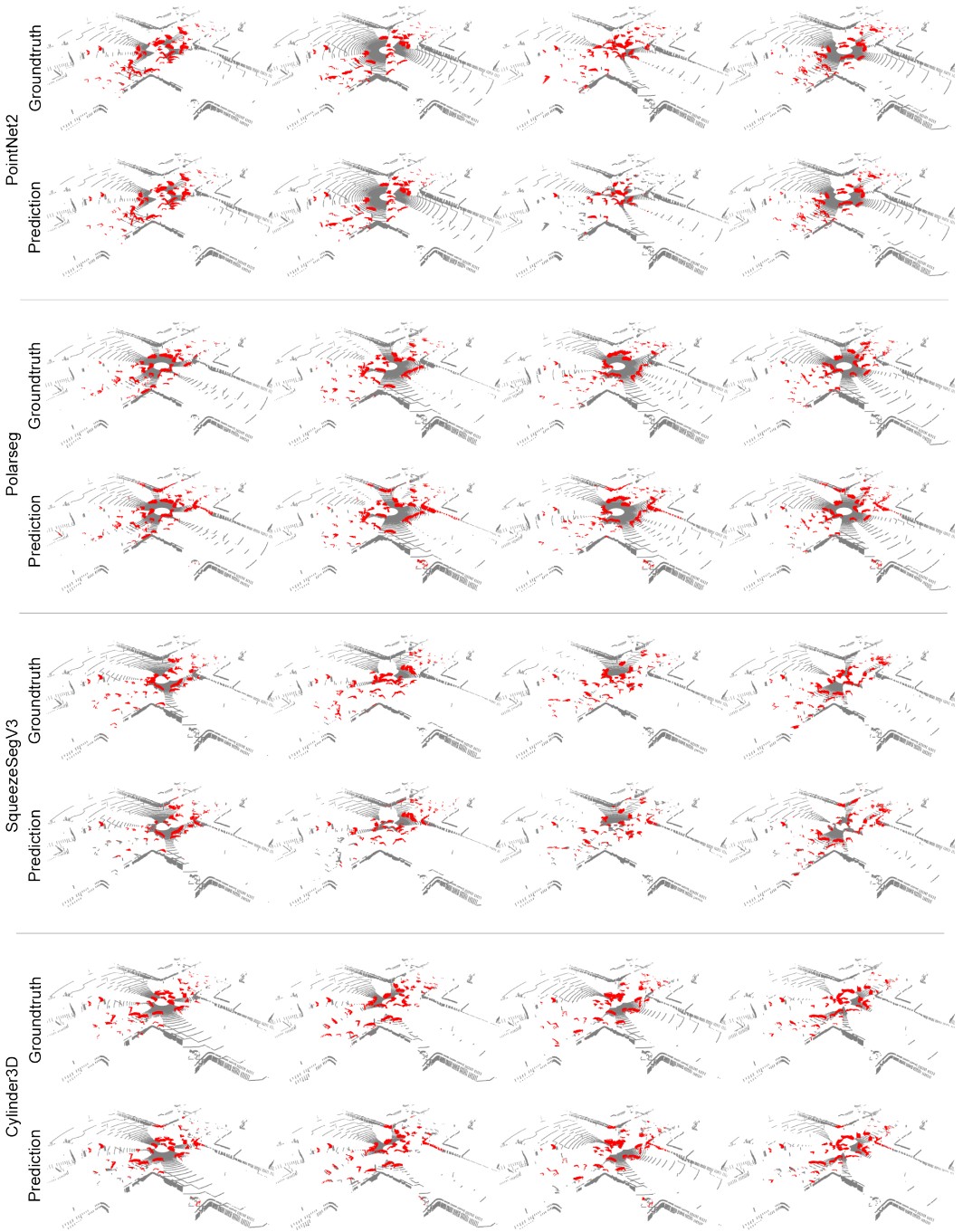

Figure 12: More results for *Pose Attack* method with the *Intersection* background.

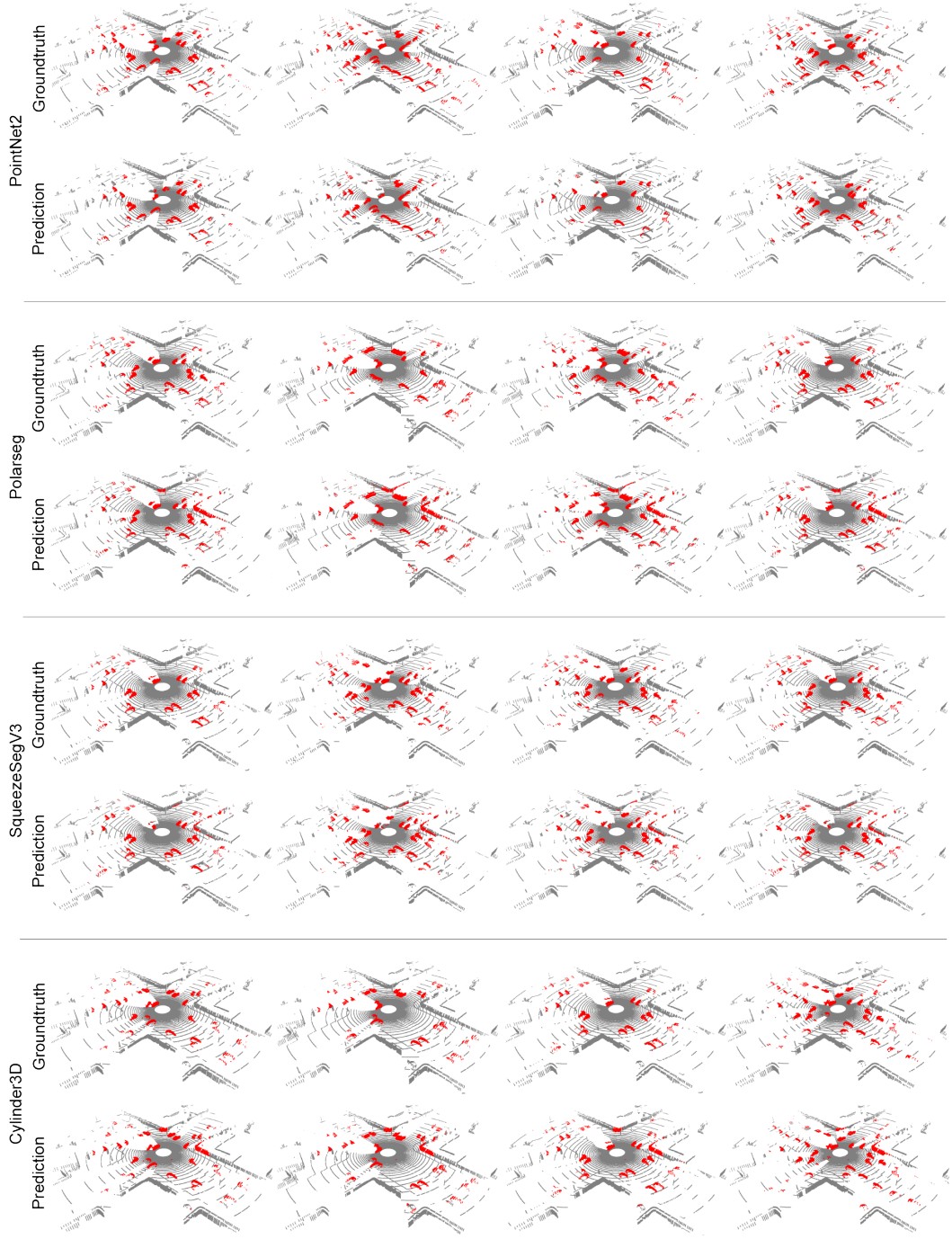

Figure 13: More results for *Scene Attack* method with the *Intersection* background.

pre-trained model provided by the authors. Since we only consider the vehicle class, we change all other labels to non-vehicle class.

### F.3 SQUEEZESEGV3

This model [71] projects the 3D point cloud to 2D range maps and extract features with 2D convolution layers from the range maps. We use the code from this repository and use the pre-trained model provided by the authors. Since we only consider the vehicle class, we change all other labels to non-vehicle class.

### F.4 CYLINDER3D

This model [75] converts the data representation from 3D Cartesian coordinate to the Polar coordinate and divides the space into blocks with a cylinder representation. We use the code from this repository and use the pre-trained model provided by the authors. Since we only consider the vehicle class, we change all other labels to non-vehicle class.

