# OpenReview forum: "Semantically Controllable Generation of Physical Scenes with Explicit Knowledge"
_ICLR.cc/2022/Conference — ICLR 2022 Submitted_

### Official Review · Reviewer_GPnn · 2021-10-31

**Correctness:** 3
**Technical Novelty And Significance:** 3
**Empirical Novelty And Significance:** 3
**Recommendation:** 6
**Confidence:** 2

**Main Review:**

This paper is quite far from my expertise so it is hard to evaluate the
originality. I have not seen this approach previously, and the literature
review supports the claimed originality.

The proposed approach does generate examples of the target scene from
random initializations, unlike most of the compared approaches that did
not have the constraints embedded. With the LIDAR scenes, the proposed approach
successfully generated more adversarial examples which also conformed
better to road traffic constraints. The adversarial examples generated
wrt one system were generally also usable as adversaries for another
system (why? what made these so effective?).

There are 2 weaknesses:
1) The paper is quite hard to follow. There is a lot of precise notational
description which could use some 'intuitive' explanation to help readers
make sense of the theory. In the case of the experiments, there is a lot of
precise detail that is missing that leads one to wonder if the claims are
a bit exaggerated. There is more detail in the appendices, and it would
help the paper to at least reference the relevant sections in the appendices.
2) There are a lot of worries about the experiments: a) why does the synthetic
target configuration have the given precise box and plate configuration,
since these are not in the constraint specification? b) The LIDAR pose attack
'cars' are at what looks like random orientations. Since the generator is
created by the authors, it is unclear why the cars could not be aligned with
the lines. c) It looks like a lot of hand-crafting is needed for constructing
the knowledge / constraint tree, and it looks like one would need a new
tree for each application. d) In the 8 cases of the LIDAR pose and scene
adversary generation, the pose approach was better in 5 of the cases, and
in the 3 cases where the proposed scene approach was better, the adversaries
were not that successful (assuming that IoU was a good measure of
adversariality).

There are a number of confusing details that could be clarified:
1) How is 'adversarial' defined in the case of the LIDAR example?
2) It is unclear what transferabilty between the Point and Scene
attack scenarios means.
3) In the case of at least 2 of the LIDAR systems, the proposed approach is
not very effective at generating adversaries. Why?
4) The paper did not clearly show how the constraints were implemented based
on the theory given in the paper, although the diagrams in the
appendices helped.
5) It was never clear what constitutes an adversary in terms of the
compared algorithms, nor why the generated LIDAR structures should be
considered an adversary.
6) Why is this? Does it not bias the experiment? "We also inject the target sce\
ne 10 times into the dataset to make
sure it accessible to all models."
7) What is the blue curve in Fig 4b?


**Summary Of The Paper:**

This paper proposes the use of tree-structured VAE as a mechanism for
encoding several forms of constraint knowledge. The VAEs create samples
of scenes that aim to conform to the constraints. A synthetic scene context
was used to demonstrate and explore the approach. A more realistic LIDAR
segmentation scenario was also explored, where the goal was to generate
realistic but adversarial scenes.


**Summary Of The Review:**

The approach seems reasonably novel, and the paper's writing could be improved. The experiments and applications are not very exciting.

---

> ### Author Response · Authors · 2021-11-14
> **Reply to Reviewer GPnn (Part 2)**
>
> **Q7: In the 8 cases of the LIDAR pose and scene adversary generation, the posed approach was better in 5 of the cases, and in the 3 cases where the proposed scene approach was better, the adversaries were not that successful (assuming that IoU was a good measure of adversariality).**
>
> In Figure 5, there are indeed 3 cases where PoseAttack outperforms our method. However, we mentioned in the analysis that the PoseAttack successes because it creates unrealistic scenes, which cannot be transferred to other victim systems (as shown in Table 2). We add this analysis to the caption of Figure 5 to make the comparison clearer.
>
> **Q8: How is 'adversarial' defined in the case of the LIDAR example?**
>
> The 'adversarial' example means examples that can reduce the performance of pointcloud segmentation algorithms (PointNet++, SqueezeSegV3, PolarSeg, Cylinder3D). We add a formal definition of adversarial attack in our experiment part.
>
> **Q9: It is unclear what transferability between the Point and Scene attack scenarios means.**
>
>
> In the revised version, we add more descriptions of transferability in our experiment part.
> Using generated adversarial examples from one system in another system is called transferability, which is very important for adversarial attacks. The reason our SceneAttack has better transferability is that the generated scenes follow basic physical laws (not overlap). With such constraints, it is much hard to attack the victim system, therefore the generated examples are “really” difficult and this difficulty is shared across different systems.
>
> **Q10: In the case of at least 2 of the LIDAR systems, the proposed approach is not very effective at generating adversaries. Why?**
>
> We assume the reviewer is referring to Figure 5. The answer could be found in Q7.
>
> **Q11: The paper did not clearly show how the constraints were implemented based on the theory given in the paper, although the diagrams in the appendices helped.**
>
> We appreciate the reviewer for pointing this out. In Section 2.2 of the paper, we provide a brief discussion of how to apply the knowledge $\mathcal{K}: x \rightarrow x'$  by searching specific nodes and changing values of properties. We also give an example of applying the constraints in Figure.2C. In the revised version, we add a new Algorithm in Section 2.2 to describe the implementation of knowledge. We hope this could help the readers better understand the algorithm.
>
> **Q12: It was never clear what constitutes an adversary in terms of the compared algorithms, nor why the generated LIDAR structures should be considered an adversary.**
>
> This question is similar to Q8, which is caused by the missing definition of adversarial attack. In our revised version, we add a formal definition of adversarial attack in our experiment part.
>
> **Q13: Why is this? Does it not bias the experiment? "We also inject the target scene 10 times into the dataset to make sure it is accessible to all models."**
>
> Firstly, we want to clarify that this will not influence the fairness of the experiment since we do this for algorithms. The reason we do this is that we want to make sure the target scene exists in the latent space (which means our goal indeed appears in our collected dataset). This operation actually makes it easier for all algorithms to find the target scene. However, we cannot always do this in realistic applications, that’s why we only do this in the toy example but not in the LiDAR adversarial attack experiment, where we cannot access the target scene.
>
> **Q14: What is the blue curve in Fig 4b?**
>
> The blue curve is the task loss (the MSE error in the first experiment and the adversarial loss in the second experiment). We show the label of the axis on the right side of the figure. We add the legend of the blue curve in our revised version.

---

> > ### Comment · Reviewer_GPnn · 2021-11-28
> > **thanks for the reply**
> >
> > Thanks for the explanations. Where they have carried over into the revised paper, they have improved it. I don't propose to change my rating of the paper. I think that the outcome depends on whether the issues in the more critical reviews have been addressed well enough. My understanding of the responses to these other issues is that they are partly addressed.

---

> ### Author Response · Authors · 2021-11-14
> **Reply to Reviewer GPnn (Part 1)**
>
> We thank the reviewer for providing detailed comments. We are also grateful that the reviewer provides concrete suggestions on improving the clarification of the paper. Since our work bridges multiple concepts from different fields, it is not easy to cover all background knowledge. The clarification suggestions of the reviewer are very clear and operable. We provided detailed answers to these questions and drafted revisions to each suggestion according to the reviewer's requests.
>
> **Q1: The adversarial examples generated wrt one system were generally also usable as adversaries for another system (why? what made these so effective?).**
>
> This is a really interesting question because the answer to this question leads to one important contribution of this paper. Using generated adversarial examples from one system in another system is called transferability, which is very important for adversarial attacks. The reason our SceneAttack has better transferability is that the generated scenes follow basic physical laws (not overlap). With such constraints, it is much harder to attack the victim system, therefore the generated examples are “really” difficult and this difficulty is shared across different systems. In contrast, examples generated by PointAttack and PoseAttack are only “spuriously” difficult, which means these examples can only attack specific victim systems (e.g. by perturbing points or making two vehicles have overlap with a special angle) and cannot attack other victims. In summary, this result shows the importance of making the generated scene follow explicit rules.
>
> **Q2: The paper is quite hard to follow. There is a lot of precise notational description which could use some 'intuitive' explanation to help readers make sense of the theory.**
>
> We thank the reviewer for providing such useful suggestions. Our T-VAE may contain some concepts that are not straightforward. We tried to illustrate the stick-breaking process with a figure in Fig.2B, but intuitive explanations for other concepts are not provided in our initial manuscript. We revise the manuscript and add more descriptions to help readers understand the high-level idea of our model.
>
> **Q3: In the case of the experiments, there is a lot of precise detail that is missing that leads one to wonder if the claims are a bit exaggerated. There is more detail in the appendices, and it would help the paper to at least reference the relevant sections in the appendices.**
>
> We thank the reviewer for providing such useful suggestions. We revise the manuscript and add the references of appendices to help readers find details.
>
> **Q4: why does the synthetic target configuration have the given precise box and plate configuration, since these are not in the constraint specification?**
>
> The task of our first toy example is to generate an image scene (rendered image) that can match a given target scene image. Therefore,  a mean square error (MSE) is used between the generated image and the target image. Minimizing this MSE encourages the generative model to move the boxes and plates to the desired positions.
>
> **Q5: The LIDAR poses attack 'cars' are at what looks like random orientations. Since the generator is created by the authors, it is unclear why the cars could not be aligned with the lines.**
>
> The pipeline of our second experiment is that we firstly use a generative model to generate poses of vehicles (poses include XYZ location and orientation), then use a LIDAR model to render these poses to a pointcloud scene. The reason why the cars have different orientations is that the generator gives us different orientations.
>
> **Q6: It looks like a lot of hand-crafting is needed for constructing the knowledge/constraint tree, and it looks like one would need a new tree for each application.**
>
> We want to clarify that the tree is not created by humans. Our T-VAE can automatically generate a tree structure neural network by composite predefined modules. Therefore, we only need to design modules but most of these modules can be shared between physical tasks. However, the knowledge and constraints need to be designed by humans. We believe this won’t be a big problem under our current setting since we can always summarize some useful information according to the downstream task but the difficult thing is how to use this knowledge (that is also our main contribution).

---

### Official Review · Reviewer_sFaB · 2021-11-02

**Correctness:** 3
**Technical Novelty And Significance:** 3
**Empirical Novelty And Significance:** 4
**Recommendation:** 6
**Confidence:** 2

**Main Review:**


Strength:
- The tree-structured generative model looks interesting, and they propose a two-stage training method to learn such a model.
- The comparison with baseline methods looks good, and the authors also show some interesting visualization results.

Weakness:
- I found this paper is a little bit hard to follow; they start from VAE and introduce the T-VAE, which is reasonable. However, aligning those proposed new functions to the basic VAE equation is not easy. I think the authors should briefly introduce the previous approaches instead of simply citing them, e.g., [49] and [55].
- The contribution of this paper is the proposed T-VAE and knowledge-guided generation. The knowledge part adopts some explicit knowledge rules. How to get those rules is not clear? Why only three rules? Also, it seems explicit rules are dependent on human experts. Can we introduce some implicit rules?
- The baseline methods are not state-of-the-art methods, and they should consider the more recent works.

**Summary Of The Paper:**

This paper proposes a method to incorporate domain knowledge into the physical scene generation process. They extend the synthetic example to realistic environments. Apart from this, they also propose the semantic point could attack against stoa segmentation methods.


**Summary Of The Review:**

The T-VAE and knowledge-guided generation proposed in this paper looks new, and the attack looks interesting. However, the current version is a little bit hard to follow, and some contributions seem incremental, for example, the proposed T-VAE compared with VAE. I can reconsider my rating if the authors point anything I was missing.

---

> ### Author Response · Authors · 2021-11-14
> **Reply to Reviewer sFaB**
>
> We thank the reviewer for providing detailed comments. Since our work bridges multiple concepts from different fields, it is not easy to cover all background knowledge. The clarification suggestions of the reviewer are very clear and operable. We provided detailed answers to these questions and drafted revisions to each suggestion according to the reviewer's requests.
>
> **Q1: I found this paper is a little bit hard to follow; Aligning those proposed new functions to the basic VAE equation is not easy. I think the authors should briefly introduce the previous approaches instead of simply citing them, e.g., [49] and [55].**
>
> We thank the reviewer for giving such a helpful suggestion. In the revised version, we provide more details about the Proximal Algorithm [49 in the paper] and the Stick-breaking process [55 in the paper].
>
> **Q2: How to get those rules is not clear? Why only three rules?**
>
> In this paper, we assume the knowledge is summarized by humans and represented by first-order logic (as shown in Definition 1 in Section 2.2). The selected rules are related to the type of scene and downstream tasks. In our toy experiment (box and plate), our target is generating a specific scene and the three rules we use are already enough to help us to achieve this goal. In our adversarial traffic scene experiment, our goal is to make all vehicles follow basic traffic rules (e.g., follow lane direction). If we consider other tasks, we could use other rules, such as following the traffic light and traffic sign.
>
> **Q3: Also, it seems explicit rules are dependent on human experts. Can we introduce some implicit rules?**
>
> Yes, our explicit rules are summarized by human experts. I assume by implicit rules the reviewer mean embeddings that are not interpretable to human (correct me if I am wrong). Usually, this kind of implicit rule is used in tasks augmented by a knowledge graph [1], where the feature is refined by knowledge. In our case, when we apply the knowledge, we just need to know the node and edge where we apply the knowledge, but the knowledge itself can be implicit rules.
>
> **Q4: The baseline methods are not state-of-the-art methods, and they should consider the more recent works.**
>
> Generating physical scenes with explicit knowledge is a relatively new area. Therefore, we don’t have off-the-shelf methods to compare. That’s why we build our own baselines with VAE and GrammarVAE. There are indeed some structural generative models used in scene generation or molecule generation. However, their method is not easy to integrate explicit knowledge. GrammarVAE is a widely used model therefore we build a baseline with this model.
>
>
> **Q5: some contributions seem incremental, for example, the proposed T-VAE compared with VAE.**
>
> VAE is a very successful and useful model for representation learning and the optimization process of our T-VAE is indeed borrowed from VAE. However, the motivation of this paper is to explore how we can use explicit knowledge to control and guide the generation of the physical scenes. In the real world, we do know something that can help our model but previously we don't know how to do it. We firstly divide the knowledge into two types then find that a tree structure is suitable to integrate these two types of knowledge. Therefore, we build a tree-structured VAE model.
>
> ---
>
> [1] Gu, Jiuxiang, et al. "Scene graph generation with external knowledge and image reconstruction." Proceedings of the IEEE/CVF Conference on Computer Vision and Pattern Recognition. 2019.

---

> > ### Comment · Reviewer_sFaB · 2021-11-29
> > **Thanks**
> >
> > Thanks for the response from the authors, now the motivation and contribution are much clear to me. However, experiments such as baselines are still limited, and I would NOT fight for this paper to be accepted.

---

### Official Review · Reviewer_gJAm · 2021-11-02

**Correctness:** 3
**Technical Novelty And Significance:** 2
**Empirical Novelty And Significance:** 2
**Recommendation:** 5
**Confidence:** 2

**Main Review:**

---
Disclaimer: I am unfamiliar with this area and can only give a very high-level review.
---

The ability to inject explicit knowledge into generative models would be very helpful for many tasks. Many generative models struggle and ignore even simple physical rules and a way to explicitly incorporate them would likely lead to better results and more data-efficient models. The authors propse to represent scenes in a tree-based manner where nodes correspond to objects and edge to relationships between objects. A VAE model is then trained to learn a structured representation of the data. After the VAE is trained the tree's characeristics can be adapted with explicit knowledge to ensure that generated scenes follow these rules.

My first question in this regard is why you focus on tree-based instead of graph-based representations? It feels to me that graph-based representations would offer more flexibility. Additionally, there is a large amount of work on graph neural networks that you could build upon. Additionally, there is a large amount of literature for graph-to-image which you could compare against.

I also don't follow the motivation for evaluating your approach on generating adversarial traffic scenes. I feel there are many other applications to evaluate the general generation capabilities. Generating adversarial attacks may not be the first thing that comes to mind for incorporating knowledge into generative models. Besides just modeling "general" scenes this could be applied to biological or chemical settigns were we have clear pre-defined rules.

As a follow-up ont this: Where do you get the knowledge from? Does the knowledge need to be hand-designed? Can the system easily scale to large rule-bases? Or is it possible to learn the rules directly from the data such that they only need to be "verified" or adjusted afterwards?

Finally, could this approach be used for editing, e.g. relationships between objects? Given that we have the tree representation, could we obtain the tree representation of a given scene, edit it, and then render the new, edited scene?

**Summary Of The Paper:**

The paper proposes a way to incorporate explicitly defined (e.g. rule based) knowledge into generative models for scene generation. Through the addition of explicit knowledge the approach can ensure that generated scenes follow specific requirements, e.g. physical rules. The approach is evaluated on a synthetic dataset and as a way to generate adversarial examples for scene based segmentation models.

**Summary Of The Review:**

I am not experienced in this domain but it feels to me that this should be more closely investigated from the view point of graph-based neural networks and graph-based image representations and rendering. The tree-based representation feels very restrictive to me. Also, I'm not sure if generating adversarial traffic scenes is the best way to evaluate this approach.

---

> ### Author Response · Authors · 2021-11-14
> **Reply to Reviewer gJAm**
>
> We thank the reviewer for providing detailed comments. Although the reviewer mentions that he or she is not familiar with this area, the high-level questions are still helpful to improve our work. Since our work bridges multiple concepts from different fields, it is not easy to cover all background knowledge. The clarification suggestions of the reviewer are very clear and operable. We provided detailed answers to these questions and drafted revisions to each suggestion according to the reviewer's requests.
>
> **Q1: why you focus on tree-based instead of graph-based representations?**
>
> The tree structure is not the only representation for scene generation but a prevalent one in recent literature [1][2][3]. In particular, as the first work to integrate domain knowledge to generative models, we select the tree structure for three reasons:
> * (1) The motivation of using trees is that objects in natural scenes, e.g., traffic, could be clustered hierarchically for better understanding and modeling. Trees naturally have the hierarchical property, where different layers represent different levels of abstractions of the scene. For example, in our T-VAE model, the high-level nodes contain the group information while leaf nodes only represent single objects.
> * (2) The hierarchical tree structure makes it easier to apply knowledge, supported by previous works in cognition literature [3].
> * (3) Compared with other scene graph models, tree structures have many existing mature approaches to improve efficiency in traversal and generation. For instance, in this paper, we consider depth-first-search traversal by using recursive neural networks.
> We add a paragraph to Section 2.1 to explain why we choose the tree structure based on the above discussion.
>
> **Q2: I also don't follow the motivation for evaluating your approach to generating adversarial traffic scenes. I feel there are many other applications to evaluate the general generation capabilities.**
>
> We agree with the reviewer that there are other applications of structure data, like indoor scene generation [4]. We select to generate adversarial traffic scenes based on the two reasons:
>
> * (1) the goal of this paper is to integrate explicit knowledge into the generative model to make generated samples follow physical law and common rules. A traffic scene is a suitable example since there are a lot of traffic rules to satisfy.
> * (2) Just satisfying rules is not the purpose of this work. To make the generated samples useful for downstream tasks, we propose this adversarial attack task to show that satisfying rules are important and useful. In contrast, generating an indoor scene may not be a good choice since we don’t have important and specific rules and there are no clear evaluation metrics to show that satisfying these rules is crucial.
>
>
> **Q3: Where do you get the knowledge from? Does the knowledge need to be hand-designed? Can the system easily scale to large rule-bases?**
>
> In this paper, we assume the knowledge is summarized by humans and represented by first-order logic (as shown in Definition 1 in Section 2.2). As for scalability, our algorithm can support complex rules as long as the rule can be represented by first-order logic. Actually, our hierarchical structure model (tree) makes it easier to apply these rules since we can divide the physical space into related regions.
>
> **Q4: Or is it possible to learn the rules directly from the data such that they only need to be "verified" or adjusted afterward?**
>
> This is a really interesting question. There is indeed some literature that tries to find rules and knowledge from data. In our case, the representation of knowledge is first-order logic. If we can develop a method to automatically extract rules in this representation, then we can directly use them in our method.
>
> **Q5: Given that we have the tree representation, could we obtain the tree representation of a given scene, edit it, and then render the new, edited scene?**
> Thanks for providing this interesting suggestion. We are able to edit an existing scene to generate new scenes. This could be used in some conditional generation tasks. Actually, a similar idea has been explored in [5] where the proposed method edits a tree of objects to change the property.
>
> ---
>
> [1] Jin, Wengong, et.al.. "Junction tree variational autoencoder for molecular graph generation."ICML 2018.
>
> [2] Mo, Kaichun, et al. "PT2PC: Learning to generate 3d point cloud shapes from part tree conditions." ECCV 2020.
>
> [3] Malcolm, George L., Iris IA Groen, and Chris I. Baker. "Making sense of real-world scenes." Trends in cognitive sciences 20.11 (2016): 843-856.
>
> [4] Li, Manyi, et al. "Grains: Generative recursive autoencoders for indoor scenes." ACM Transactions on Graphics (TOG) 38.2 (2019): 1-16.
>
> [5] Mo, Kaichun, et al. "StructEdit: Learning structural shape variations." CVPR 2020.

---

### Official Review · Reviewer_EqAi · 2021-11-03

**Correctness:** 2
**Technical Novelty And Significance:** 2
**Empirical Novelty And Significance:** 2
**Recommendation:** 5
**Confidence:** 4

**Main Review:**


## Strengths
* Training a model to generate scene layouts while enforcing explicit constraints during training is novel. However, training a neural network while enforcing explicit constraints on the output has been studied before [Ref 1].

## Weaknesses
* "GVAE leverages knowledge by integrating rules during the training stage; however, it cannot explicitly integrate semantic knowledge during the generation." -- Disagree, rules are explicit knowledge about the scenes and can certainly be incorporated in the similar fashion as  [Ref 2] as long as those are simple co-occurrence based. However, the advantages of the current model over the sg-vae seem to be the followings: (1) the current method could model any constraints of the objects (possible higher order constraints)  whereas sg-vae could model only co-occurrences of different objects, and (2) the optimization-based formulation of the current method enforces a soft-constraints where as the grammar-based structure of sg-vae enforces hard-constraints on the object co-occurrences.

* "When combining the three knowledge, even from a random initialization, our T-VAE can finally find the target scene, leading to a small error in Table. 1." --- The external knowledge is adding explicit constraints in the scene generation. Those constraints reduce the search space and that leads to a better minimum? That being said, the main point here to discuss is how one could incorporate the external knowledge for the current scene generation task. This work does so by fine-tuning a t-vae while enforcing the constraints on the scene layout. An interesting evaluation however would be the comparison w.r.t. The following baselines (1) T-vae + imposing the constraint during test time, (2) current model (trained with explicit constraints, ie external knowledge) without explicit constraints during testing time. Since the constraints were imposed during the training time of the current method, the network might already know how to impose the constraints.  Also, the external knowledge constraints were not employed in the baselines Direct Search (DS) and VAE.

* The scene attack in the current method is not convincing. To me the adversarial attack is something where we modify the image / scene by a small amount (invisible to the naked eye) to deceive an existing trained model. Now, if a trained model performs poorly on a synthesized scene then the scene could be bad. It would be more convincing if the attack changes only a tiny amount of the 3d point cloud and still could  fool a trained model.

* The synthetic scene layouts utilized in this work are rather simple. An experiment on the suncg datasets comparing against a stronger baseline [Ref. 3] would be good.


## References
* [Ref. 1] Gould, Stephen, Richard Hartley, and Dylan Campbell. "Deep declarative networks: A new hope." arXiv preprint arXiv:1909.04866 (2019).
* [Ref. 2] Purkait, Pulak, Christopher Zach, and Ian Reid. "SG-VAE: Scene Grammar Variational Autoencoder to generate new indoor scenes." European Conference on Computer Vision. Springer, Cham, 2020.
* [Ref 3] Li, Manyi, et al. "Grains: Generative recursive autoencoders for indoor scenes." ACM Transactions on Graphics (TOG) 38.2 (2019): 1-16.

**Summary Of The Paper:**

The current method proposes a scene generation technique that models explicit knowledge by imposing semantic rules on the objects in the scene. It generates meaningful scene layouts and shown to be perform better than some simple baselines.

**Summary Of The Review:**

### Summary
* A novel formulation and a sound approach to generate novel scenes
* Weak evaluation -- very simple synthetic dataset, unconvincing attack
* Weak baselines -- no comparison against s.o.t.a scene layout generation methods

---

> ### Author Response · Authors · 2021-11-14
> **Reply to Reviewer EqAi (Part 2)**
>
>
> **Q5: Also, the external knowledge constraints were not employed in the baselines Direct Search (DS) and VAE.**
>
> Due to the lack of structure, it is impossible to apply knowledge constraints to DS and VAE as same as T-VAE. However, in both methods, we need to fix the dimension of x to represent 10 objects (2 plates and 8 boxes), which already leaks information about the correct number and types of objects. However, we don’t know which object is close to which, so we cannot apply the complex knowledge to describe relations.
>
> **Q6: The scene attack in the current method is not convincing. To me, the adversarial attack is something where we modify the image/scene by a small amount (invisible to the naked eye) to deceive an existing trained model.**
>
> We apologize for missing a clear definition of the adversarial attack we use. In our revised version, we modify the related works section and add more discussion of semantically adversarial attacks in the experiment setting. In general, the default adversarial attack in the computer vision area is regularized by the $L_p$ norm. The target is generating images within a $L_p$ norm bound to reduce the victim’s performance. However, the term adversarial attack is not restricted to this setting. Some existing works [1][2][3][4][5] focus on semantic or physical adversarial attack, where the constraint is semantic constraint instead of $L_p$ norm.
> A very related adversarial example to our setting is adding an object on top of a vehicle to make the vehicle disappear in detection algorithms [2][3].
>
> **Q7: The synthetic scene layouts utilized in this work are rather simple. An experiment on the suncg datasets comparing against a stronger baseline would be good**
>
> We appreciate the reviewer for providing this potential experiment setting. We agree that generating an indoor scene is also one possible application of our method. However, in this paper, we focus on generating a traffic scene where knowledge is crucial to make the generation work. Our first toy example is only used to investigate the knowledge-guided generation. Although it looks very similar to the indoor scene generation task, we are not trying to beat other methods in this setting.
>
>
> ---
>
> [1] Liu, Hsueh-Ti Derek, et al. "Beyond pixel norm-balls: Parametric adversaries using an analytically differentiable renderer." arXiv preprint arXiv:1808.02651 (2018).
>
> [2] Tu, James, et al. "Physically realizable adversarial examples for lidar object detection." Proceedings of the IEEE/CVF Conference on Computer Vision and Pattern Recognition. 2020.
>
> [3] Abdelfattah, Mazen, et al. "Towards Universal Physical Attacks On Cascaded Camera-Lidar 3D Object Detection Models." arXiv preprint arXiv:2101.10747 (2021).
>
> [4] Jain, Lakshya, et al. "Analyzing and Improving Neural Networks by Generating Semantic Counterexamples through Differentiable Rendering." arXiv preprint arXiv:1910.00727 (2019).
>
> [5] Alcorn, Michael A., et al. "Strike (with) a pose: Neural networks are easily fooled by strange poses of familiar objects." Proceedings of the IEEE/CVF Conference on Computer Vision and Pattern Recognition. 2019.

---

> > ### Comment · Reviewer_EqAi · 2021-12-01
> > **Thanks**
> >
> > I thank the authors for carefully answering my concerns. However, the target application (adversarial attack) is not convincing in this scenario and hence I cannot assess the results. The paper is about "scene generation" and I was expeting to see results on the qualitty of the scene generation. The method is evaluated for adversarial attacks which is not established in this setting and I am not conviced that this is the best way to evaluate the current method.

---

> ### Author Response · Authors · 2021-11-14
> **Reply to Reviewer EqAi (Part 1)**
>
> We thank the reviewer for providing detailed comments. We are also grateful that the reviewer provides concrete suggestions on improving the clarification of the paper. Since our work bridges multiple concepts from different fields, it is not easy to cover all background knowledge. The clarification suggestions of the reviewer are very clear and operable. We provided detailed answers to these questions and drafted revisions to each suggestion according to the reviewer's requests.
>
> **Q1: Training a neural network while enforcing explicit constraints on the output has been studied before.**
>
> We thank the reviewer for providing this relevant reference. We add a more detailed discussion between our method and previous literature. Injecting explicit constraints into neural networks is a widely studied area. Previous work focuses on giving a general framework to combine explicit knowledge with NN. The main contribution of our paper is representing the physical scene with a tree structure, where we can naturally integrate node-level and edge-level knowledge to control the relationship between objects. Therefore, we are targeting a more specific problem but it still has broad application in the real world.
>
> **Q2: Disagree, rules are explicit knowledge about the scenes and can certainly be incorporated in a similar fashion as [Ref 2] as long as those are simple co-occurrence based.**
>
> We thank the reviewer for pointing out this important point. It is indeed possible to integrate simple knowledge into GVAE during the generation. We removed our statement in Section 3.1 and add some discussion of the comparison between our method and GVAE in evaluation results.
>
> **Q3: The external knowledge is adding explicit constraints in the scene generation. Those constraints reduce the search space and that leads to a better minimum?**
>
> Yes, this is the most possible explanation, and our experiment results shown in Figure 4(c) support this explanation.
>
> **Q4: An interesting evaluation however would be the comparison w.r.t. the following baselines (1) T-vae + imposing the constraint during test time, (2) current model (trained with explicit constraints, i.e. external knowledge) without explicit constraints during testing time.**
>
> Let’s firstly clarify the setting of our experiment: (1) During the training stage, we don’t use the explicit constraints in any methods (DS, VAE, GVAE, T-VAE, T-VAE w/ SCG). Since GVAE and T-VAE are structured models, we do need to create grammars for GAVE and input tree-structured data for T-VAE. However, during the training stage, the datasets contain very different scenes (different number of objects, different colors, and different positions), therefore we don’t use any explicit knowledge for all methods. (2) Then, during the testing stage (optimization), only T-VAE w/ SCG uses the explicit knowledge, whereas other methods only minimize the objective of the task $\mathcal{L}_t(x)$.
> The reason we use this setting is that it is not easy to directly apply knowledge to a generative model. The structure of T-VAE makes it possible to apply complex explicit knowledge. The comparison between T-VAE and T-VAE w/ SCG is designed to show the power of explicit knowledge. We are glad to discuss more possible settings of the experiment with the reviewer since we believe this is important to show our main contribution.

---

### Author Response · Authors · 2021-11-22
**Discussions with Reviewers**

We thank all reviewers for their time and valuable reviews. We hope that we have covered the reviewers’ concerns. Since the current discussion period is closing soon, it would be helpful to know the reviewers' comments on our responses. We also appreciate it if the reviewers raise any remaining concerns.

Thanks!

---

### Decision · Program_Chairs · 2022-01-20

**Decision:**

Reject

**Comment:**

This paper presents a semantically controllable generative framework by integrating explicit knowledge. In particular, a tree-structured generative model is proposed based on knowledge categorization. Reviewers raised concerns about technical details, experiments, and missing references. In the revised paper, the authors provided more justifications and clarifications, such as the definition of adversarial attack. During the discussion, reviewers agreed that the previous concerns have been partially addressed, but there are still concerns on experiments, e.g., more recent work should be considered as baselines.

Overall, I recommend to reject this paper. I encourage the authors to take the review feedback into account and submit a future version to another venue.